# Distinct genomic routes underlie transitions to specialised symbiotic lifestyles in deep-sea annelid worms

Giacomo Moggioli[1], Balig Panossian [1], Yanan Sun[2,3,4], Daniel Thiel[5], Francisco M. Martín-Zamora [1], Martin Tran[1], Alexander M. Clifford[6], Shana K. Goffredi [7], Nadezhda Rimskaya-Korsakova[8], Gáspár Jékely [5], Martin Tresguerres [6], Pei-Yuan Qian [2,4], Jian-Wen Qiu [3,4], Greg W. Rouse [6], Lee M. Henry [1] ✉ & José M. Martín-Durán [1] ✉

Bacterial symbioses allow annelids to colonise extreme ecological niches, such as hydrothermal vents and whale falls. Yet, the genetic principles sustaining these symbioses remain unclear. Here, we show that different genomic adaptations underpin the symbioses of phylogenetically related annelids with distinct nutritional strategies. Genome compaction and extensive gene losses distinguish the heterotrophic symbiosis of the bone-eating worm *Osedax frankpressi* from the chemoautotrophic symbiosis of deep-sea Vestimentifera. *Osedax*'s endosymbionts complement many of the host's metabolic deficiencies, including the loss of pathways to recycle nitrogen and synthesise some amino acids. *Osedax*'s endosymbionts possess the glyoxylate cycle, which could allow more efficient catabolism of bone-derived nutrients and the production of carbohydrates from fatty acids. Unlike in most Vestimentifera, innate immunity genes are reduced in *O. frankpressi*, which, however, has an expansion of matrix metalloproteases to digest collagen. Our study supports that distinct nutritional interactions influence host genome evolution differently in highly specialised symbioses.

Symbioses have shaped life on Earth, from the origin of the eukaryotic cell to the formation of biodiversity hotspots such as coral reefs[1,2]. Animal chemosynthetic symbioses, where bacteria convert inorganic compounds to organic matter, are ubiquitous in marine habitats[3] and fuel some of the most productive communities, such as those around hydrothermal vents[4]. Siboglinid worms (Annelida) often dominate deep-sea chemosynthetic environments through symbioses with environmentally acquired bacteria[5,6] that adults harbour within a specialised organ called a trophosome[7]. Despite their ecological

importance, the host's genetic traits sustaining these symbioses have only been studied in Vestimentifera[8–10], one of the four main lineages in Siboglinidae (Fig. 1a). The genomes of *Lamellibrachia luymesi*[8], *Paraescarpia echinospica*[9], *Riftia pachyptila*[10] and *Ridgeia piscesae*[11] have revealed a complex molecular interplay between Vestimentifera and their endosymbionts to fulfil their nutritional demands[12]. For example, the hosts have lost genes involved in amino acid biosynthesis[8,10,11] and carbohydrate catabolism[9] but expanded gene families involved in nutrient transport[8], gas exchange[8–10,13,14], innate immunity[9,11] and

[1]School of Biological and Behavioural Sciences, Queen Mary University of London, Mile End Road, E1 4NS London, UK. [2]Department of Ocean Science, The Hong Kong University of Science and Technology, Hong Kong, China. [3]Department of Biology, Hong Kong Baptist University, Hong Kong, China. [4]Southern Marine Science and Engineering Guangdong Laboratory (Guangzhou), Guangzhou 511458, China. [5]Living Systems Institute, University of Exeter, Exeter, UK. [6]Scripps Institution of Oceanography, University of California, San Diego, La Jolla, CA 92093, USA. [7]Department of Biology, Occidental College, Los Angeles, LA, USA. [8]Friedrich Schiller University Jena, Faculty of Biological Sciences, Institute of Zoology and Evolutionary Research, Erbertstr. 1, 07743 Jena, Germany. ✉e-mail: l.henry@qmul.ac.uk; chema.martin@qmul.ac.uk

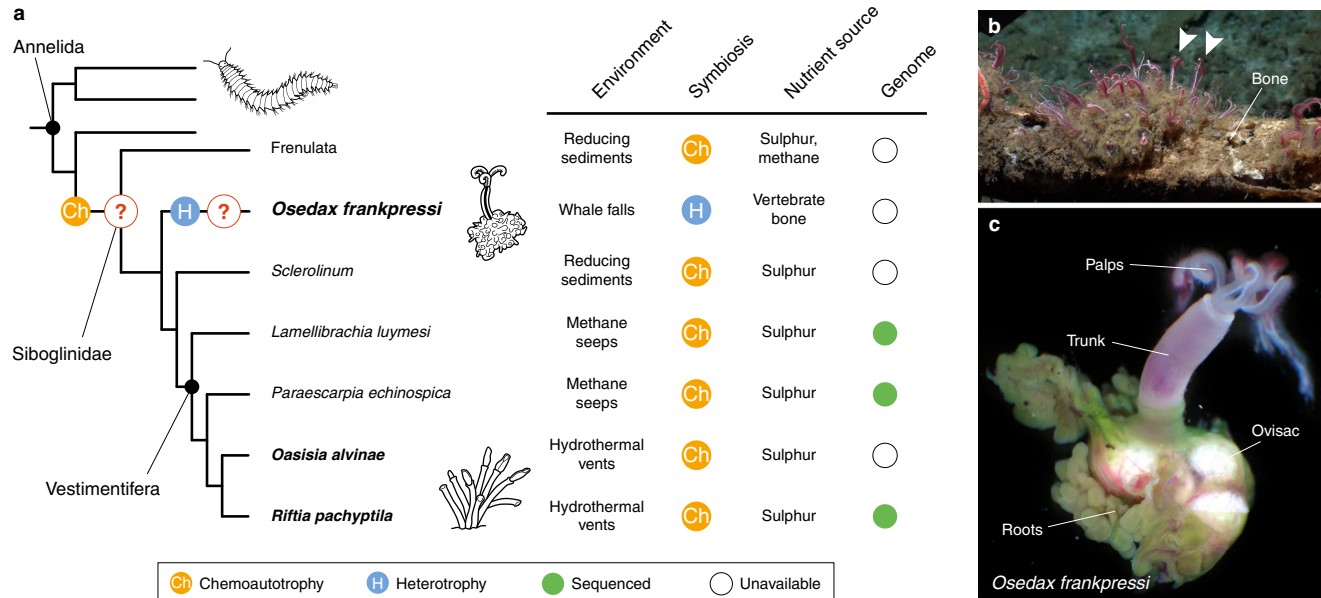

**Fig. 1 | Siboglinidae is a symbiotic annelid group. a** Siboglinidae is a diverse clade of annelid worms that evolved chemosynthetic symbioses (left side). There are four main lineages within Siboglinidae, namely Frenulata, *Osedax*, *Sclerolinum* and Vestimentifera. Chemolithoautotrophy occurs in Frenulata, *Sclerolinum* and Vestimentifera, which associate with gammaproteobacteria that employ sulphur or methane to produce organic compounds in an array of marine ecosystems, from reducing sediments to methane seeps and hydrothermal vents (right side of the panel). Differently, *Osedax* worms (e.g., *O. frankpressi*; **b**, **c**) have secondarily evolved a heterotrophic association with *Oceanospirillales* to exploit decaying vertebrate bones. The genomic basis for the evolution of these nutritional symbioses in Siboglinidae is unclear (question marks on the left) because genomic information only exists for Vestimentifera hosts (green circles on the right). The species herein studied are highlighted in boldface. **b**, **c** Photographs of *O. frankpressi* in a whale bone (**b**; arrowheads point to *O. frankpressi*) and a mature female adult (**c**). *O. frankpressi* settles and colonises decaying vertebrate bones (**b**). There, the posterior part of the body becomes stably infected with environmentally acquired *Oceanospirillales* bacteria. This body part (the so-called roots) harbours the bacteria and grows to penetrate the bone, dissolving the organic components. These nutrients are absorbed and transported towards the bacteriocytes containing the endosymbionts, which will proliferate and act as food for the worm. Anterior to the root tissue there are the reproductive ovisacs and the head bears two pairs of palps.

lysosomal digestion[8–10,15]. On the other hand, there is genomic information for the endosymbionts of most major clades of Siboglinidae, including Vestimentifera, *Osedax* and Frenulata[16–18]. The endosymbionts of Vestimentifera and Frenulata are mixotrophs[19] and show a diverse metabolic repertoire for energy production (e.g., the reductive tricarboxylic acid cycle in the endosymbionts of Vestimentifera) and nutrient biosynthesis that complements the metabolic deficiencies of, at least, the vestimentiferan host[16,17]. In addition, an increase in the genetic repertoire to infect and evade the host's immunity[16–18], transport nutrients[18] and metabolise nitrogen compounds[16,17] is common in endosymbionts of Siboglinidae. Notably, many of these genetic changes also occur in other distantly related chemosymbiotic animals, including bivalves[20], gastropods[21], and the clitellate annelid *Olavius algarvensis*[22]. Therefore, disparate animal groups have convergently evolved similar genetic mechanisms to sustain different chemosynthetic symbioses in marine ecosystems.

Within Siboglinidae, the marine *Osedax* annelids have evolved a unique endosymbiosis[23–27] with heterotrophic bacteria in the order *Oceanospirillales*[18,24,28–30] (Fig. 1a) that allows them to obtain nutrients from bones of dead animals lying on the ocean's floor (Fig. 1b). While *Osedax* shares some morphological features with other siboglinids[31], including the lack of a gut, mouth and anus, *Osedax* contains bacteriocytes concentrated in the subepidermal connective tissue of the lower trunk that grows directly into the bone[24,28] (Fig. 1c). This amorphous tissue, referred to as "roots", expresses high levels of V-type H⁺-ATPase and carbonic anhydrase[32], indicating acid is used to dissolve the bone matrix to access collagen and lipids, which are then absorbed across the root epithelium. Enzymatic[28,29] and transcriptomic data[33] support this theory by showing that the roots of *Osedax* express many proteases and solute carrier transporters that are thought to be involved in bone degradation and nutrient absorption, perhaps with the aid of the endosymbionts[18]. However, it is currently unclear whether the specialised heterotrophic symbiosis of *Osedax* is based on homologous genetic traits to those discovered in Vestimentifera and other chemoautotrophic invertebrates or if it relies on unique genomic adaptations. Untangling the molecular mechanisms behind this remarkable symbiosis is, therefore, central to understanding the evolution of *Osedax* and Siboglinidae, as well as the ecological principles and succession of bone-eating communities[34].

In this study, we sequenced the genome of *Osedax frankpressi* Rouse, Goffredi & Vrijenhoek, 2004[24], as well as that of two vent-dwelling Vestimentifera, *Oasisia alvinae* Jones, 1985 and *Riftia pachyptila* Jones, 1981, and compared them with nearly 40 eukaryote and prokaryote genomes to better understand the genomic changes leading to these distinct symbiotic lifestyles. In contrast to Vestimentifera, we found that *O. frankpressi* has a small AT-rich genome with a reduced gene repertoire. Gene families typically expanded in chemosymbiotic hosts, such as innate immunity components, are reduced in *O. frankpressi*. Instead, the *Osedax-Oceanospirillales* symbiosis has unique genomic adaptations for bone digestion, including the loss of biosynthetic pathways of amino acids that are abundant in vertebrate bones in the host, the presence of the glyoxylate cycle in the endosymbiont that could allow the production of carbohydrates from the lipids present in vertebrate bones, and the expansion of matrix metalloproteases in the host that could aid in bone digestion. Together, our findings demonstrate that different genomic principles sustain the nutritional symbioses of *Osedax* and Vestimentifera, providing critical insight into the genetic and metabolic changes that have enabled symbiotic siboglinids to colonise diverse nutrient-imbalanced feeding niches.

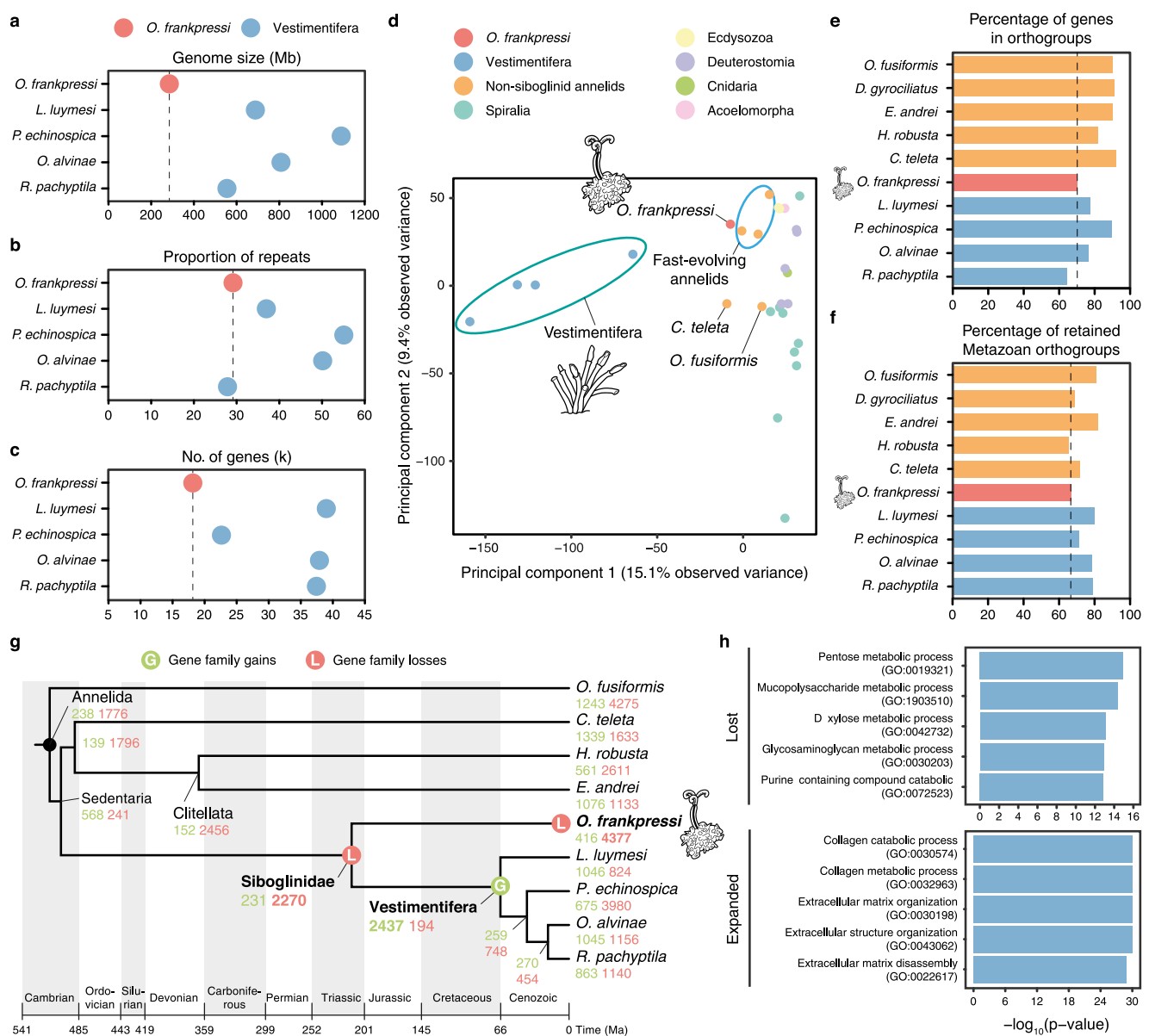

**Fig. 2 | *Osedax* and Vestimentifera exhibit distinct genome evolutionary trends. a–c** Plots comparing genome size (**a**), repeat content (**b**) and number of genes (**c**) between *O. frankpressi* and the four Vestimentifera with sequenced genomes. *Osedax frankpressi* has a smaller genome, with less genes but relatively similar repeat content. **d** Principal component analyses of the gene content of 28 metazoan genomes show that differently from symbiotic bivalves and gastropods, the gene content of Vestimentifera and *O. frankpressi* differs from slow-evolving asymbiotic species (as represented by *Owenia fusiformis* and *C. teleta*). While Vestimentifera has a unique gene content, *O. frankpressi* is like other fast-evolving annelid lineages. *Osedax frankpressi* is amongst the annelids with less genes in gene families and less retained ancestral metazoan genes. **g** Patterns of gene family gains (in green) and loss (in red) during the evolution of Annelida under a consensus tree topology[31] and a consensus of published molecular dates[8,9]. A major event of gene loss is common to all Siboglinidae. While *O. frankpressi* continued experiencing high rates of gene loss, a major event of gene innovation is common to all Vestimentifera. **h** Top five enriched gene ontology terms (Biological Process) for gene families lost (top) and expanded (bottom) in *O. frankpressi*. While *O. frankpressi* has further lost genes involved in metabolism (e.g., carbohydrate metabolism), genes involved in collagen and extracellular matrix degradation are expanded. *P*-values were derived from upper-tail Fisher's exact tests.

## Results

### The genome of *O. frankpressi*

To identify genomic signatures that could inform the genetic and physiological basis of the heterotrophic symbiosis in *Osedax*, we used long PacBio reads and short Illumina reads to assemble the genome of *O. frankpressi*[24] (Supplementary Table 1). We also sequenced the genomes of two Vestimentifera from hydrothermal vents, *Oasisia alvinae* and *R. pachyptila* (Supplementary Fig. 1), complementing previous genome sequencing efforts[8–10]. We generated almost entirely haploid draft assemblies (Supplementary Fig. 2a–d), which included the

circularised endosymbiont genomes of *O. frankpressi* and *Oasisia alvinae* and several epibionts associated with *O. frankpressi* (Supplementary Fig. 2h–j; Supplementary Table 2). Consistent with *k*-mer-based analyses (Supplementary Fig. 2e–g), previously reported genome size estimation for *Oasisia alvinae*[35], and a recent genome assembly of *R. pachyptila*[10], the assembled genomes for *O. frankpressi*, *Oasisia alvinae* and *R. pachyptila* span 285 Mb (1,185 scaffolds with an N50 of 426 Kb), 808 Mb (642 scaffolds with an N50 of 2.975 Mb) and 554 Mb (918 scaffolds with an N50 of 1.424 Mb) after removal of bacterial contigs, respectively (Fig. 2a; Supplementary Fig. 2k).

The genome assemblies for *Oasisia alvinae* and *R. pachyptila* shows high completeness (96.9% and 95.6% BUSCO presence, respectively; Supplementary Fig. 2l; Supplementary Table 3). The assembly for *O. frankpressi* appeared to have lower completeness (80.1% BUSCO presence; Supplementary Fig. 2l). However, 95.62% and 97.77% of the de novo assembled transcripts from the body and root tissue mapped to the genome assembly of *O. frankpressi*, respectively. Accordingly, BUSCO completeness increased to a final score of 96.23% after gene annotation (Supplementary Fig. 2l) and manual curation (26 out of the 62 missing BUSCO could be manually annotated; Supplementary Data 1). Together, this suggests that the fast rates of molecular evolution in coding sequences observed in *Osedax* worms[36] are likely responsible for the relatively low initial, assembly-based BUSCO completeness in the genome of *O. frankpressi*.

Although the genome of *O. frankpressi* is ~50–75% smaller than the sequenced genomes and estimated genome sizes of Vestimentifera[8–10,35] (Fig. 2a), the fraction of simple repeats and transposable elements in *O. frankpressi* (29.16%) is comparable to that of the vestimentiferan *R. pachyptila* (27.87%) and asymbiotic annelids with similar genome sizes (Fig. 2b; Supplementary Fig. 3a). As in Vestimentifera, the repeat landscape in *O. frankpressi* shows signs of expansions (Supplementary Fig. 3b), unlike in asymbiotic annelids with slow rates of molecular evolution[37,38]. Combining transcriptomic evidence (Supplementary Table 1) with ab initio gene prediction (Supplementary Fig. 2a), we functionally annotated 37,777 and 38,179 protein-coding transcripts in *Oasisia alvinae* and *R. pachyptila*, respectively (Supplementary Fig. 2k), which have a similar number of genes to other Vestimentifera and asymbiotic annelids[8,37,38]. The number of genes annotated in our assembly for *R. pachyptila* is higher than in a previous report[10] (Supplementary Fig. 4a). Still, both annotations and assemblies are broadly equivalent (Supplementary Fig. 4b–d). Unlike Vestimentifera, *O. frankpressi* has a smaller repertoire of 18,657 transcripts (Fig. 2c), comparable to that of the miniaturised *Dimorphilus gyrociliatus*[36], another annelid species with a compact genome and a streamlined gene set (14,203 genes). Therefore, *O. frankpressi* has the smallest genome of all sequenced siboglinids. Given the number of genes in genomes of asymbiotic annelids, gene loss rather than removal of repeat content seems to account for the genome size difference between these two lineages of Siboglinidae.

### Gene gains and losses shape the evolution of Siboglinidae

To investigate gene content evolution between major lineages of Siboglinidae, we first reconstructed the gene families of 28 highly complete metazoan genomes, including seven symbiotic annelid and molluscan lineages (Supplementary Data 2). This taxonomic sampling provides sufficient resolution to infer the time of origin of each gene family while minimising potential biases in orthology inference in fast-evolving species[39]. A principal component analysis of the number of orthologs per gene family in the 28 species clustered the symbiotic molluscs *Bathymodiolus platifrons*[20] and *Gigantopelta aegis*[21] with their asymbiotic bivalve and gastropod relatives, respectively (Supplementary Fig. 5a). However, the four Vestimentifera species are markedly differentiated from the other annelid and animal genomes, and *O. frankpressi* is closer to heterotrophic annelids with fast rates of molecular evolution and divergent gene repertoires, such as the leech *Helobdella robusta* and the earthworm *Eisenia andrei*—which also harbour bacterial symbionts[40–42]—and the marine worm *D. gyrociliatus* (Fig. 2d; Supplementary Fig. 5a). Indeed, after *R. pachyptila*, *O. frankpressi* is the annelid with the second lowest percentage of genes assigned to gene families (Fig. 2e) and has only retained a fraction of ancestral metazoan gene families comparable to more rapidly evolving annelids such as *H. robusta* and *D. gyrociliatus* (Fig. 2f). Therefore, unlike symbiotic molluscs, the evolution of nutritional symbioses in

Siboglinidae correlates with divergent host gene repertoires compared to their asymbiotic annelid counterparts.

To identify and characterise the evolutionary events underpinning the divergent gene repertoires of Siboglinidae, we reconstructed the patterns of gene family evolution in those 28 metazoan genomes under a consensus tree topology (Supplementary Fig. 5b). Vestimentifera and *O. frankpressi* share a major gene loss event involving 2270 gene families of mainly ancient origins (61.23% of the lost families originated before Metazoa and the Bilateria/Nephrozoa ancestor) (Fig. 2g) and enriched in Gene Ontology (GO) terms associated with metabolism (Supplementary Fig. 5c). This loss thus coincides with the evolution of nutritional symbioses in the last common ancestor of Siboglinidae. A high rate of gene loss continued in the *O. frankpressi* lineage (Fig. 2g), which ultimately accounts for its reduced gene repertoire and primarily affected genes associated with carbohydrate and nitrogen metabolism (Fig. 2h; Supplementary Fig. 5d). Notably, Vestimentifera experienced an event of gene family expansion in its last common ancestor (2,437 gene families), mainly affecting genes related to immunity, cell communication, and response to stimuli[9] (Fig. 2g; Supplementary Fig. 5e). However, high lineage-specific rates of gene loss also occur in some Vestimentifera[10,11], as in *O. frankpressi* (Fig. 2g). Compared to Vestimentifera, *O. frankpressi* has had few gene family gains (Supplementary Fig. 5b) but has experienced a large expansion of gene families associated with extracellular matrix remodelling and degradation (e.g., collagen degrading proteases; Fig. 2h; Supplementary Fig. 5f) in agreement with previous transcriptomic observations[33]. Altogether, our findings indicate that the evolution of symbiosis in *Osedax* and Vestimentifera relies on different host gene repertoires, one sculptured predominantly through gene loss (in *O. frankpressi*) and another through ancestral gene gains followed by varying, species-specific rates of gene loss (in Vestimentifera)[8–10] (Fig. 2g).

### The different genomic traits of Siboglinidae endosymbionts

To investigate the genetic and functional contribution of the endosymbionts to the nutritional symbioses of Siboglinid worms, we used our PacBio long-read data to assemble the genomes of the primary endosymbionts of *O. frankpressi* (Rs1 ribotype; Genome Taxonomy Database accession number Rs1 sp000416275) (Fig. 3a; Supplementary Data 3), and *Oasisia alvinae* (Supplementary Fig. 6; Supplementary Data 4), as well as several epibionts associated with *Osedax*[43] (Supplementary Table 2). The circularised assembly of the endosymbionts of *O. frankpressi* improved the previously published genome[18], revealing 95 new functional genes that provide additional insights into its symbiosis (Supplementary Data 5). Compared to deep-sea free-living relatives, the *O. frankpressi* endosymbiont has a genome enriched in metabolic genes for protein secretion systems, carbohydrate metabolism, and coenzyme and amino acid biosynthesis (Supplementary Data 6b). This includes additional virulence factors, such as multiple complete copies of Type 5a, 5b, and 6i secretion system pathways (Supplementary Data 6c) that are important for modulating interactions with other bacteria and eukaryotic hosts. *Neptunomonas japonica*, a close relative of the *Oceanospirillales* endosymbionts recovered from marine sediments near a whale fall, has many of the same metabolic capabilities of the endosymbionts; however, it lacks the additional secretion systems[44]. The Type 5a and Type 5b secretion systems are also largely absent in the endosymbionts of Vestimentifera (Supplementary Data 7g). This increase in virulence factors may reflect that *Oceanospirillales* repeatedly infect the roots of *Osedax* as it grows through bone material, unlike the trophosome of Vestimentifera, which is colonised early during host development[7,45]. In addition, all Siboglinidae endosymbionts contain numerous genes encoding eukaryote-like protein domains, which, interestingly, tend to be host-lineage-specific (Supplementary Data 8). Eukaryote-like proteins

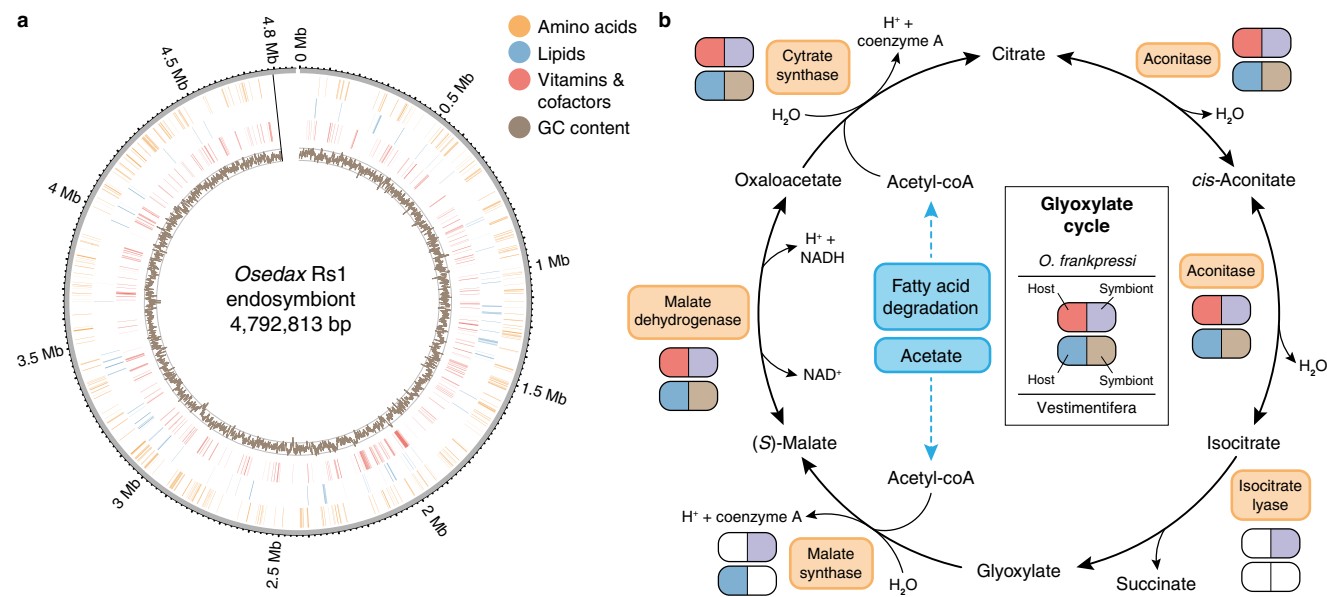

**Fig. 3 | *Osedax* and its endosymbiont reconstruct the glyoxylate cycle.**
**a** Circular schematic representation of the genome of *Osedax* endosymbiont Rs1, assembled into a single contig. The plot shows the genomic location of genes involved in amino acid, lipid and vitamin/cofactor metabolism (in orange, blue and red, respectively) and the GC content (inner circle; brown colour).
**b** *Oceanospirillales* endosymbionts possess the glyoxylate cycle, a metabolic

pathway that can produce oxaloacetate, which can serve as the precursor to synthesise carbohydrates from the oxidation of fatty acids. This metabolic pathway could thus contribute to synthesising glucose in a diet (the bone) that is naturally poor in carbohydrates. Notably, this molecular and metabolic interaction does not occur between Vestimentifera and its symbionts because the host and microbes lack the enzyme isocitrate lyase.

modulate important processes in many symbioses, including extracellular secretions, cell binding and colonisation[12,46,47]. Therefore, the specificity of the endosymbionts' eukaryote-like proteins in the different lineages of Siboglinidae suggests they may be important for host and clade-specific annelid-symbiont communication, as shown in *Riftia*[12].

*Osedax frankpressi*'s endosymbionts and those of Vestimentifera and Frenulata shared a broadly similar repertoire of genes involved in core cellular processes (Supplementary Data 7a). However, as we may expect for a heterotrophic microbe, the *Oceanospirillales* endosymbionts have significantly more genes involved in the metabolism and uptake of amino acids, coenzymes, lipids, and carbohydrates (Supplementary Fig. 7a–c; Supplementary Data 7a, e, f). This includes several complete pathways to convert oxaloacetate into ribose 5-phosphate that can be used in the biosynthesis of nucleotides and histidine, the Entner-Doudoroff and De Ley-Doudoroff pathways to catabolise carbohydrates, and multiple sugar, amino acid and oligopeptide ATP-binding transporters (Supplementary Data 7). In addition, the endosymbionts of *O. frankpressi* can produce all essential amino acids (including methionine and threonine) and vitamin B6, unlike Vestimentifera and Frenulata endosymbionts, as well as vitamin B2, which Vestimentifera symbionts cannot make (Fig. 4a; Supplementary Data 7d, e). Notably, the B2 pathway was considered missing in the previous draft genome of the *Oceanospirillales* endosymbiont[18], but it is present in ours. As in some of the bacteria comprising the microbiome of degrading bones[48], *O. frankpressi*'s endosymbionts can catabolise hydroxyproline, one of the most abundant amino acids in collagen[49,50], but lacks a secreted M9 peptidase to cleave extracellular collagen (Supplementary Data 3). Finally, all endosymbionts of Vestimentifera are enriched in genes involved in chemosynthesis, most of which are absent in the heterotroph endosymbionts of *Osedax* (Supplementary Fig. 7c). Taken together, our results confirm and expand previous genomics efforts on the *Oceanospirillales* endosymbionts[18], further demonstrating that Siboglinidae has partnered with metabolically versatile microbes that are suited to sustain symbioses with eukaryotes in diverse environments.

## Metabolic adaptations for bone digestion

Vertebrate bones are nutrient-imbalanced food sources enriched in lipids and proteins and deficient in carbohydrates[50]. Given the reduced gene repertoire of the host (Fig. 2c) and the metabolic versatility of its endosymbionts[18] (Supplementary Data 7), we next explored potential molecular and metabolic interactions that could facilitate the nutritional specialisation of *O. frankpressi*. Combining highly sensitive profile hidden Markov Models sequence similarity searches with KEGG and COG functional annotations, we reconstructed all metabolic routes in *O. frankpressi* and its endosymbionts and the published genomes of Vestimentifera and their respective endosymbionts. *Osedax frankpressi* and Vestimentifera have similar metabolic capabilities to produce and process lipids (Supplementary Data 9). However, *O. frankpressi* has six incomplete pathways for carbohydrate metabolism that are intact in Vestimentifera and other asymbiotic annelids (Supplementary Data 9), consistent with the loss of gene families involved in carbohydrate metabolism (Fig. 2h). In one case (UDP-N-acetyl-D-glucosamine biosynthesis), the endosymbiont possesses the enzymes that would complement the losses in *O. frankpressi* (Supplementary Data 7c, Supplementary Data 9). Notably, the endosymbionts, unlike the host, lack the enzymes glycogen synthase and glycogen phosphorylase and, therefore, cannot produce glycogen (Supplementary Data 7) and possess the enzymes to complete the glyoxylate cycle (Fig. 3b, Supplementary Data 7), which allows the production of glucose from the catabolism of fatty acids and acetate[51–54]. This metabolic pathway does not occur in Vestimentifera because both the host and endosymbiont lack an isocitrate lyase (Fig. 3b). Therefore, the glyoxylate cycle may play a role in the metabolic interaction of *Osedax* and its endosymbionts by collectively converting bone lipids into carbohydrates, which are often nearly absent in bones[50]. Although *Osedax* appears to use wax esters to store energy[29], the fat content of bones varies widely, and *Osedax* can grow in dentin (Greg W. Rouse and Shana K. Goffredi, personal observation), where lipids are a minor component. Functional studies are thus warranted to assess the nutritional and physiological relevance of this metabolic pathway in *Osedax* and under different nutritional sources.

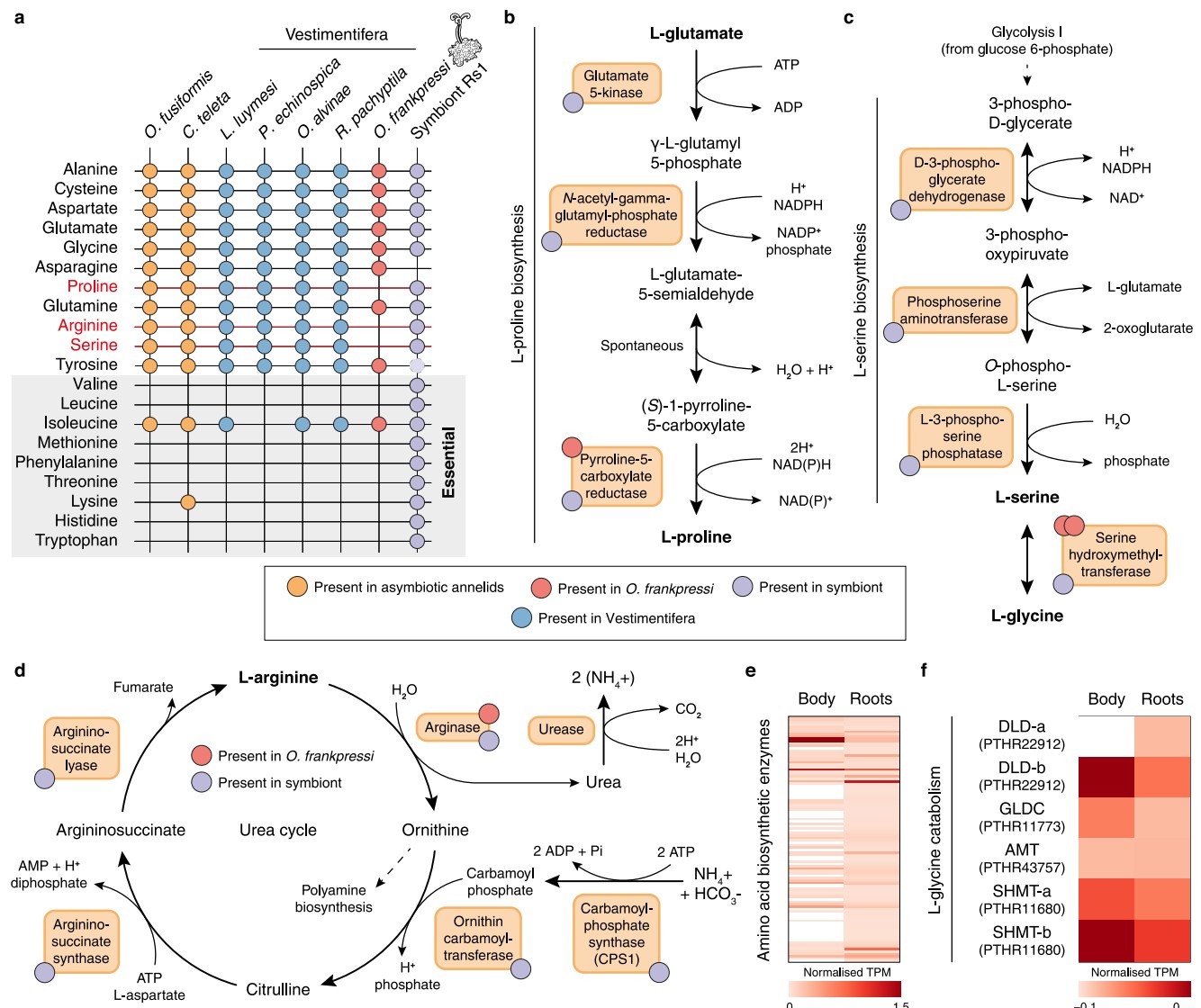

**Fig. 4 | *Osedax*'s metabolic adaptations to bone digestion. a** Summary table of the presence (filled circles) and absence (empty crosses) of amino acid biosynthetic pathways in seven annelid genomes and *O. frankpressi* endosymbiont (symbiont Rs1). While Vestimentifera and asymbiotic annelids can synthesise all amino acids that are non-essential and conditional for humans, *O. frankpressi* shows incomplete pathways to synthesise proline, arginine, and serine (in red). Some of these amino acids are abundant in the bone (e.g., proline) and all can be produced by the symbiont (tyrosine biosynthetic pathway is truncated in the symbiont; dotted and lighter circle). **b–d** Schematic representation (as in MetaCyc database) of the biosynthetic pathways for proline (**b**), serine (**c**) and arginine (**d**) indicating with red

and violet circles the enzymes present in *O. frankpressi* and its endosymbiont, respectively. *Osedax frankpressi* cannot produce serine from glycolytic metabolites but can either produce serine from collagen-derived glycine or take it from the diet. In addition, *O. frankpressi* can only convert arginine into ornithine, producing urea as a result. **e, f** Heatmaps of normalised mRNA expression levels for amino acid biosynthetic enzymes (**e**) and glycine catabolising enzymes (**f**) in the body and roots of *O. frankpressi*. Biosynthetic enzymes (**e**), including the two copies of serine hydroxymethyltransferase (SHMT-a and SHMT-b) that convert glycine into serine, are more expressed in the roots than in the body of *O. frankpressi*. Source data for (**e, f**) are provided as a Source Data file.

Proteins, predominantly collagen[50], are the core organic component of bone. Collagen is rich in proline/hydroxyproline and glycine[49], and thus its amino acid composition is also imbalanced. Consistent with previous genomic analyses[8–10], Vestimentifera and asymbiotic annelids (*Owenia fusiformis* and *C. teleta*) can produce all non-essential and conditionally essential amino acids. However, *O. frankpressi* cannot synthesise the amino acids proline, serine, and arginine (which are non-essential or conditional for mammals), but its endosymbionts can (Fig. 4a). Indeed, only one enzyme (pyrroline-5-carboxylate reductase) of the proline biosynthetic pathway remains, which is expressed at similar levels in the roots and the rest of the body, unlike most amino acid biosynthetic enzymes that are enriched in roots (Fig. 4b, e). Similarly, the entire pathway to synthesise serine from intermediates of glycolysis is missing in *O. frankpressi* (Fig. 4c). However,

*O. frankpressi* (like other annelids) has an intact glycine cleavage system (Fig. 4f), which would favour the conversion of collagen-derived glycine into serine through serine hydroxymethyltransferase[55]. The two copies of this enzyme are highly expressed throughout *O. frankpressi* (Fig. 4f) and could provide an additional source of serine on top of those offered by the diet and endosymbionts. Therefore, *O. frankpressi* shows genomic-inferred metabolic adaptations to its unique bone-eating diet in its gene complement, which differs from the more intact metabolic repertoire of Vestimentifera and other asymbiotic annelids[12].

The catabolism of amino acids produces ammonia, a compound that can be toxic but can also serve as a substrate for amino acid biosynthesis by both animals and bacteria. Most aquatic organisms excrete excess ammonia into the water, but a few aquatic animals and

most air-breathing vertebrates shuttle ammonia into the urea cycle leading to urea production[56]. *Osedax frankpressi* lacks four urea cycle enzymes and only possesses arginase (Fig. 4d). Interestingly, the urea cycle is also incomplete in the leech *Poecilobdella granulosa*[57], another symbiotic heterotrophic annelid with a protein-rich diet that excretes ammonia as a waste product. In *O. frankpressi*, the lack of CPS1 is especially significant because this enzyme is the rate-limiting step that mediates the entry of ammonia into the urea cycle; in fact, CPS1 genetic deficiency in humans leads to episodic toxic ammonia levels in the blood ("hyperammonemia")[58]. However, *O. frankpressi* additionally lacks urease; therefore, this enzyme is not available to convert ammonia (and carbon dioxide) into urea, thus ensuring elevated internal ammonia levels. The only enzyme present in the urea cycle of *O. frankpressi* is arginase, which catalyses the interconversion of arginine—which the worm likely obtains from bone-derived collagen and the endosymbionts (Fig. 4a)—into ornithine and urea. Although the urea produced by this pathway can be expected to be negligible for ammonia homeostasis, the ornithine may generate putrescine and other polyamines essential for multiple cellular functions[59]. Therefore, the amino acid-rich diet and lack of a urea cycle almost certainly imply chronic hyperammonemia in *Osedax*. This would favour amino acid biosynthesis by both *Osedax* and their endosymbionts; however, further functional experiments are needed to test this scenario.

## Lineage-specific expansions of matrix metalloproteinases

As a core component of vertebrate bones, collagen is poised to be an essential nutrient for *Osedax*[28,29,32] and the bone-associated microbiome[48]. Accordingly, transcriptomic analyses uncovered numerous metalloproteases expressed in the root tissue of *O. japonicus*[33]. Our gene family evolutionary analyses also showed that genes involved in collagen catabolism and extracellular matrix organisation are expanded in the genome of *O. frankpressi* (Fig. 2h; Supplementary Fig. 5f). Amongst these expanded families, genes annotated as matrix metalloproteases (MMPs) are the greatest fraction (24.3%). To investigate how MMPs diversified in *O. frankpressi*, we extracted the reconstructed gene families and functional annotations of symbiotic and asymbiotic annelids to identify sequences containing a metallopeptidase domain (InterPro accession IPR006026). We then reconstructed a phylogeny of the metallopeptidase genes using maximum likelihood and Bayesian approaches (Fig. 5a; Supplementary Fig. 8, 9). Our analyses recovered all previously described classes of vertebrate MMPs with high statistical support (bootstrap node support >80%) (Fig. 5a, highlighted in green) and discovered eight new highly supported invertebrate-specific classes of MMPs, labelled A to H (Fig. 5a, highlighted in blue). In addition, we identified two *Osedax*-specific large clades of MMPs, which we referred to as MMP-Os1 and MMP-Os2 (Fig. 5a, highlighted in red). The *Osedax*-specific expansions are more closely related to invertebrate than to vertebrate collagenases, supporting previous enzymatic observations that suggested generic proteolysis rather than an actual collagenase activity in *Osedax* worms[28]. The majority of MMPs belonging to MMP-Os1 (37.5%) had a metallopeptidase domain combined with a C-terminal hemopexin-like repeats domain (IPR018487) thought to facilitate binding to other components of the extracellular matrix[60] (Fig. 5b; Supplementary Fig. 10). As observed with the 12 MMPs reported in *O. japonicus*[33], all but two of the 63 MMPs found in *O. frankpressi* are more highly expressed in root tissue than in the rest of the body (Fig. 5c). At least 43 out of 63 (68.25%) have a signal peptide. This suggests the MMPs are excreted across the root-bone interface—similar to bone-degrading osteoclast cells of vertebrate animals[61]—allowing *Osedax* to digest bone-derived collagen extracellularly and absorb the resulting nutrients through the root epithelium for direct consumption, transport to the endosymbiont for further catabolism[18,32,33], or both. Therefore, the large expansion of MMPs in an otherwise reduced genome is a unique trait of *Osedax* that may be related to their ability to exploit bones from

diverse vertebrates, hence collagens with different amino acid sequences and protease-cleavage sites.

## Divergence in innate immunity repertoire

Establishing stable and specific host-bacterial associations involves innate immunity genes, which are expanded in some Vestimentifera[8,9] (Supplementary Fig. 5e) and other symbiotic oligochaetes[22]. To identify the immune gene repertoire in *O. frankpressi*, we investigated the reconstructed gene families for innate immune pattern recognition receptors corresponding to six major classes, namely lectins, peptidoglycan recognition proteins, Toll-like receptors, scavenger receptors, bactericidal permeability-increasing proteins, and NOD-like receptors[62]. Compared to asymbiotic annelids (i.e., *Owenia fusiformis* and *C. teleta*) and Vestimentifera, *O. frankpressi* has fewer immunity genes in all considered classes (Fig. 6; Supplementary Table 7; Supplementary Data 11–17). This includes a smaller repertoire of Toll-like receptors, which are expanded in some species of Vestimentifera[8,9], and the loss of galectin and a NOD-like receptor, which is a family of cytosolic immune receptors that recognises and triggers inflammatory responses to bacterial pathogens[63] that are also largely expanded in Vestimentifera[9] (Supplementary Table 7; Supplementary Data 12–15). Notably, there is no clear association between the expression levels of the different classes of pattern recognition receptors and the body regions and tissues of Siboglinidae. Yet, a C-type lectin is highly expressed in the root tissue of *O. frankpressi* (Fig. 6). Our findings indicate that *O. frankpressi* and Vestimentifera have different innate immune complements that are simplified in the former and generally expanded in the latter. Further research in Frenulata and *Sclerolinum* will inform whether this divergence in the repertoire of innate immune genes may underpin the evolution of a novel symbiotic association with *Oceanospirillales* bacteria in *Osedax* worms.

## A conserved developmental toolkit in Siboglinidae

In addition to lacking a gut, Vestimentifera and *Osedax* also lack eyes and any other sensory structure in their most anterior region, the prostomium[64]. Yet unlike other annelids with unusual body plans, such as the leech *H. robusta*[38], the genomes of Vestimentifera contain a complete developmental toolkit[9,10]. To investigate genes involved in body patterning and organogenesis in the reduced gene set of *O. frankpressi*, we first focused on the repertoire of G protein-coupled receptors (GPCRs; Supplementary Data 18), a large family of evolutionarily related membrane receptors involved in an array of developmental, sensory, and hormonal processes[65,66]. All siboglinids show a conserved repertoire of GPCRs of class B (secretins), C (metabotropic glutamate receptors) and F (frizzled and smoothened receptors) (Supplementary Fig. 11b–d). However, Siboglinidae has a more divergent complement of rhodopsin-like receptors (class A), with five expanded clusters, one specific to *O. frankpressi* (Supplementary Fig. 11a, highlighted in pink). Notably, *O. frankpressi* and Vestimentifera have lost four GPCR families, including opsins (Supplementary Fig. 11a, highlighted in grey), suggesting an ancestral loss of light perception to these groups in parallel to the colonisation of light-deprived deep marine environments.

The bulk of the body of Siboglinidae has only two segments and the posterior end (i.e., the opisthosoma), which is often multisegmented, is lacking in *Osedax*[31,64]. Nevertheless, the complement of *Hox* genes—a conserved family of transcription factors that define a molecular code throughout the many trunk segments in Annelida[37,67]—is largely conserved in Vestimentifera, only missing the gene *Antennapedia (Antp)*[9,10]. *Osedax frankpressi* has a similar *Hox* gene repertoire, and thus the loss of *Antp* might have occurred in the last common ancestor of Siboglinidae (Supplementary Figs. 12a, 13a). Indeed, the number and complement of transcription factors involved in animal development are comparable in *O. frankpressi*, Vestimentifera and asymbiotic annelids, except for Basic Leucine Zipper Domain

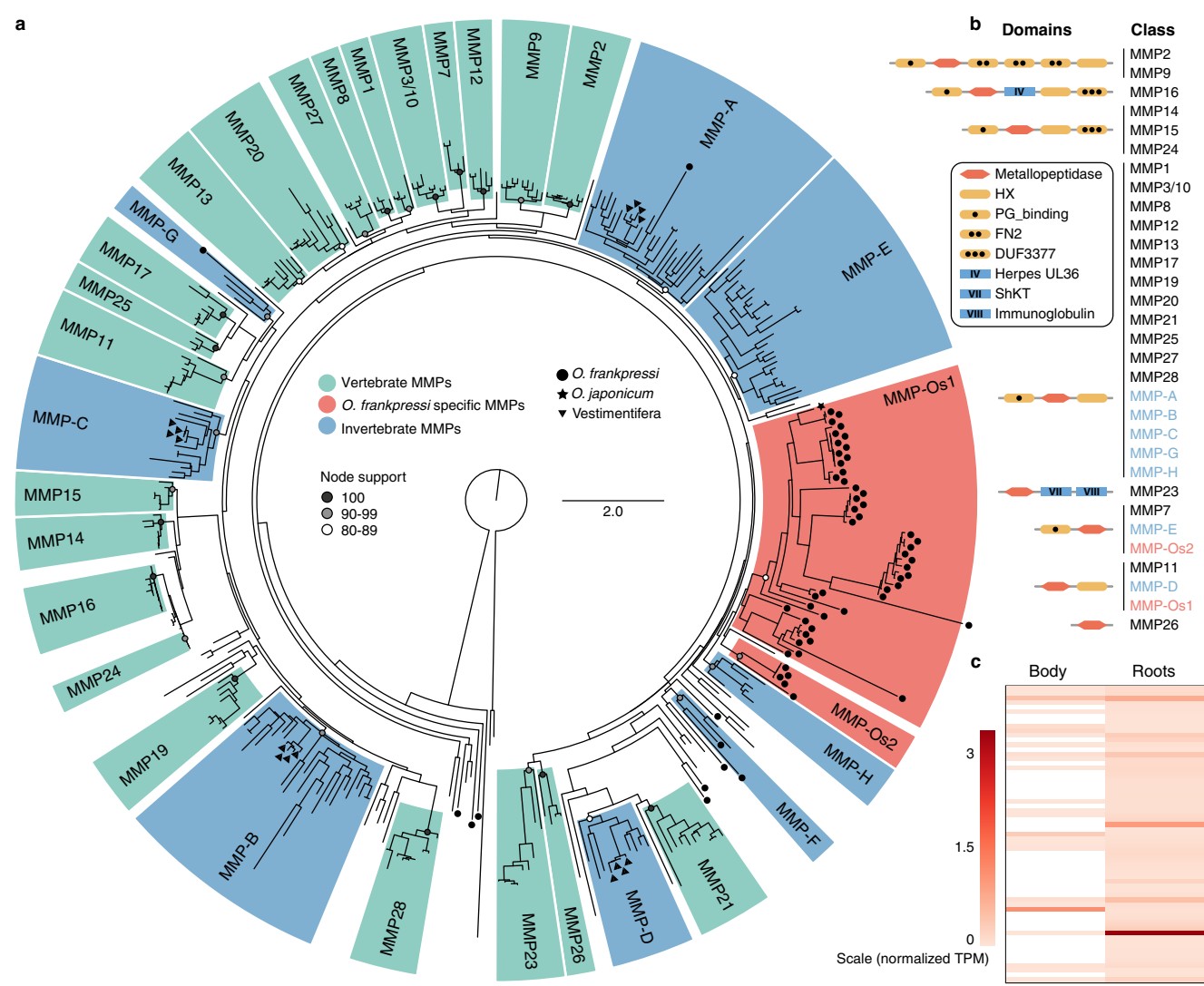

**Fig. 5 | Matrix metalloproteases experienced lineage-specific expansions in *O. frankpressi*. a** Phylogenetic reconstruction of animal matrix metalloproteases (MMPs) based on the metallopeptidase domain. Tree topology is based on maximum likelihood reconstruction and node bootstrap support for each major class is colour coded (white circles show an 80–89 bootstrap support; grey circles indicate a 90–99 bootstrap values and black dots highlight fully supported nodes). Vertebrate-specific MMP classes are highlighted in green and named according to existing literature[161]. New monophyletic clades of invertebrate MMPs are in blue and named from A to H. *Osedax frankpressi* experienced two independent expansions of MMPs, shown in red and named as MMP-Os1 and MMP-Os2. **b** Schematic drawings of the protein domain composition of the different MMP classes recovered in (**a**). For each class, only the most abundant domain architecture is shown. A complete characterisation of the domain composition of MMPs is in Supplementary Fig. 10. Drawings are not to scale. **c** Heatmap of normalised expression levels of MMPs in the body and roots of *O. frankpressi*. Most MMPs show higher expression levels in the roots than in the body. Source data for (**c**) are provided as a Source Data file.

containing proteins (bZIP; PF00170) and zinc finger transcription factors (C2H2-Zn; PF00096), which are reduced (Supplementary Fig. 13b; Supplementary Data 19), as well as certain specific classes, such as the *ParaHox* genes (Supplementary Fig. 12a). Similarly, *O. frankpressi* retains all major developmental signalling pathways, yet it has a lower number of Notch containing proteins (Supplementary Fig. 13c, d) and a simplified repertoire of signalling ligands (Supplementary Figs. 12b, 14, 15), as also observed in the miniaturised annelid *D. gyrociliatus*[36]. Therefore, *O. frankpressi* and Vestimentifera show a similar and generally conserved developmental toolkit, suggesting that changes in gene regulation rather than deviations in the gene complement underpin the development of the divergent adult morphology of Siboglinidae after symbiont acquisition.

**Species-specific repertoires of DNA damage repair mechanisms**
Changes in the machinery that repair DNA damage can cause biases in the GC composition of the genome[68,69], and such changes have been associated with genome compaction and gene loss in animals[70]. *Osedax frankpressi* has an AT-rich genome (29.08% GC content versus ~41% observed in Vestimentifera; Supplementary Fig. 2k, m) and unlike other annelids, it has three major DNA repair pathways that are largely incomplete, namely the base excision repair, the non-homologous end joining, and the Fanconi anaemia DNA repair pathway (Supplementary Figs. 12c, 16). The base excision repair pathway corrects DNA damage from base lesions caused by deamination, oxidation and methylation, and is thought to increase GC to AT base transitions when impaired[71]. The lack of the non-homologous end joining pathway—the most common mechanism to repair double-strand DNA breaks[72]—triggers the error-prone microhomology-mediated end joining pathway, which is intact in *O. frankpressi* and all other annelids but causes microdeletions[73] (Supplementary Fig. 16f; Supplementary Data 20). Therefore, the loss of genes involved in the repair of double-strand DNA breaks and chemical base modifications might underpin the reduction in genome size and GC content observed in *O. frankpressi* in

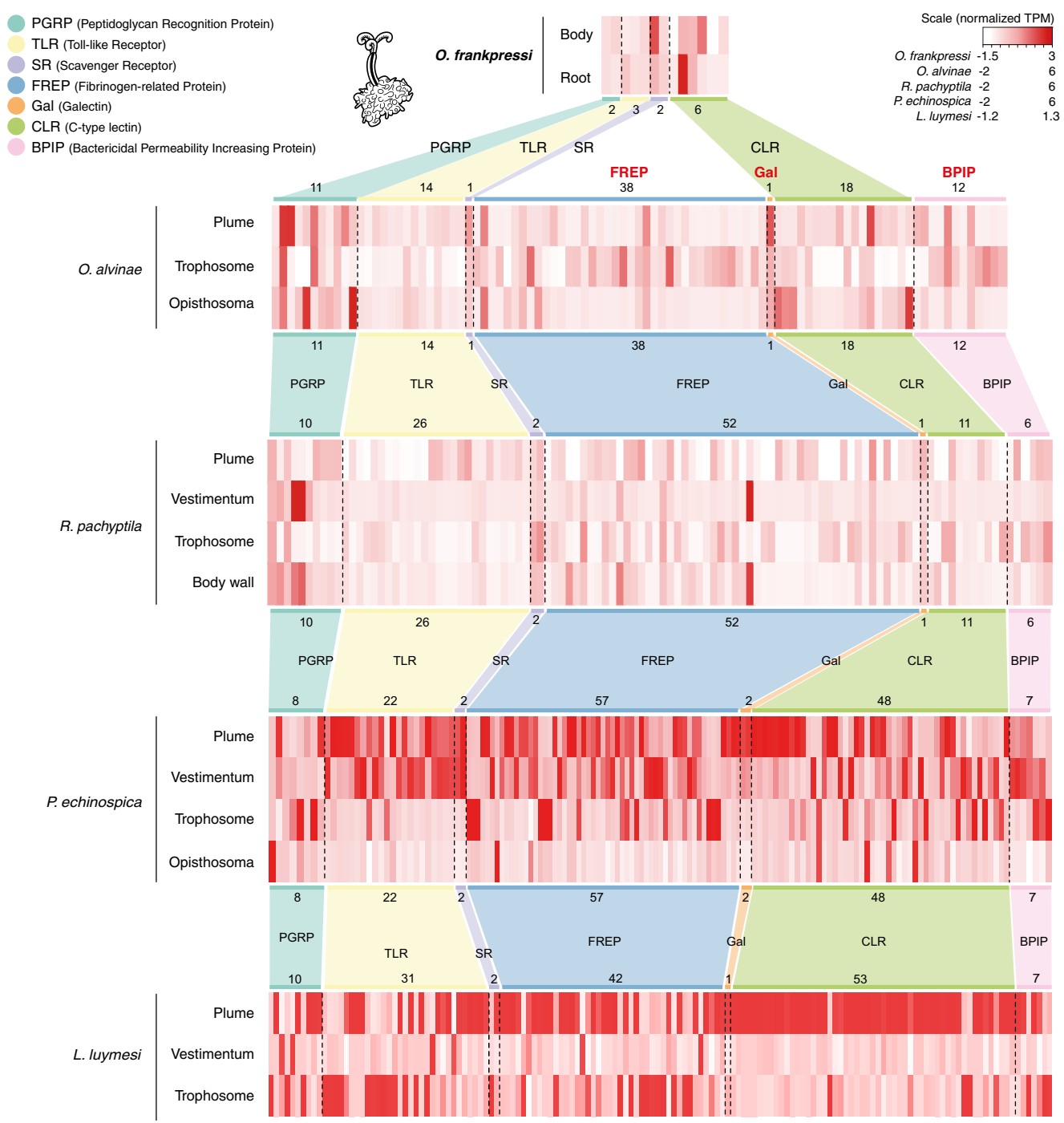

**Fig. 6 | *O. frankpressi* has a reduced innate immune gene repertoire compared to Vestimentifera.** Heatmaps of tissue-specific normalised gene expression of innate immune genes in four species of Siboglinidae, including *O. frankpressi* (top) and the Vestimentifera *Oasisia alvinae*, *R. pachyptila* and *P. echinospica*. While Vestimentifera have relatively similar repertoires of innate immune genes, *O. frankpressi* has a much-reduced complement (Supplementary Table 7). Notably,

innate immune genes do not show a clear tissue-specific expression within or among species of Siboglinidae. The immune repertoire and gene expression values for *P. echinospica* and *L. luymesi* is based on previously published genome resources[8, 9]. Source data for *O. frankpressi, Oasisia alvinae* and *R. pachyptila* are provided as a Source Data file.

comparison with Vestimentifera, thus differing from other annelids with reduced genomes, such as *D. gyrociliatus*, whose genome eroded without changes in DNA repair pathways[36].

## Discussion

Our data reveal additional evidence on the genetic interactions and co-dependencies of animal hosts and bacterial symbionts that have enabled distinct symbiotic lifestyles, including the exploitation of

sunken vertebrate bones as a food source (Fig. 7a). Our analyses of the genomes of *Oasisia alvinae* and *R. pachyptila* confirm what was previously reported for other species of Vestimentifera[8,9] and *R. pachyptila* itself[10] and support that broadly similar genomic adaptations underpin the different symbioses of Vestimentifera, even between species occupying distinct environments, such as hydrothermal vents and methane seeps. However, compared to Vestimentifera, *O. frankpressi* shows a fast evolving[36], divergent gene

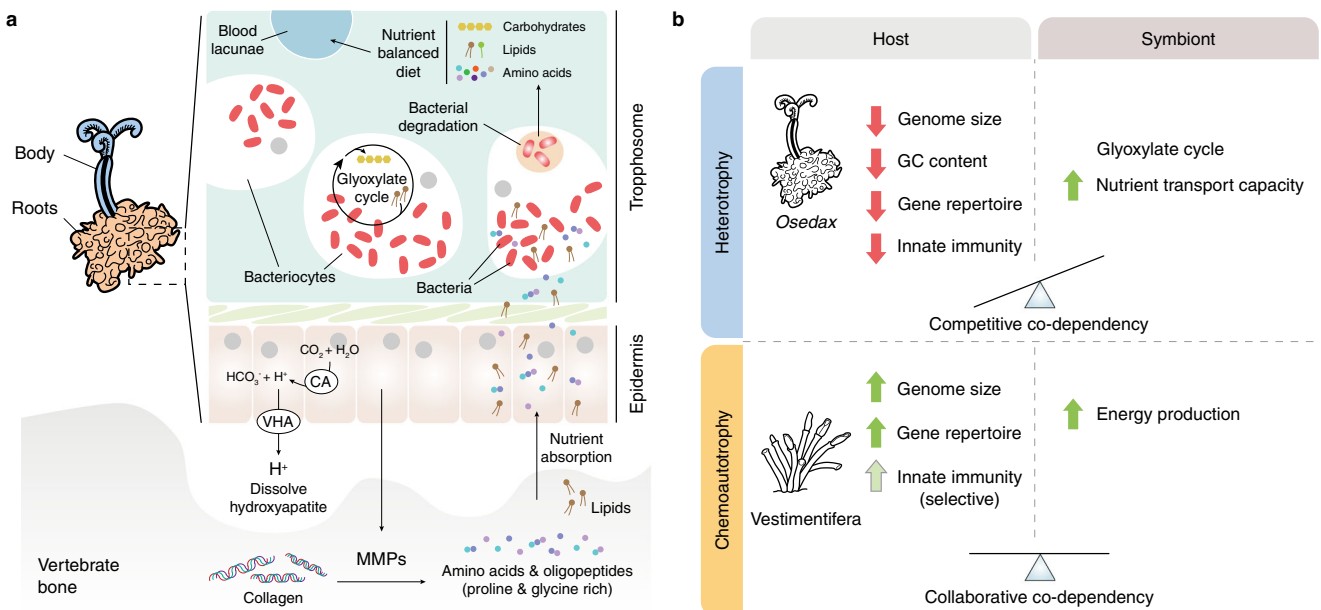

**Fig. 7 | The genomic basis and evolution of different nutritional symbioses in Siboglinidae. a** Schematic drawing of the metabolic interaction for bone digestion between *Osedax* and its endosymbiont (red kidney-shaped ovals), which are harboured in the trophosome inside bacteriocytes. The root epidermis secretes acid to dissolve the inorganic component of the bone (via carbonic anhydrase, CA, and V-type H+-ATPase, VHA) and matrix metalloproteases (MMPs) that break collagen, one of the most abundant organic components of the bone, into amino acids and oligopeptides, which are rich in proline and glycine. These amino acids and the lipidic content of the bone are absorbed by the epidermis and used either directly by *Osedax* or transported to bacteriocytes. The host and endosymbiont cooperate to generate carbohydrates (in low amounts in bone) from the oxidation of fatty acids (abundant in bone and roots) through the glyoxylate cycle, most likely inside the bacteriocytes. Ultimately, these interactions transform the originally unbalanced diet into complex and diverse macronutrients, which are then taken directly

or after the digestion of the bacteria by the host. **b** *Osedax* and Vestimentifera broadly show different genomic traits. *Osedax* has a small, AT-rich genome, with many gene losses and a reduced immune repertoire. Vestimentifera tends to show larger genomes, with a more extensive gene complement and richer innate immunity (although there is inter-specific variation for these traits, highlighted with a lighter green arrow). We hypothesise that the different nutritional relationships between the hosts and symbionts in these two groups might explain, at least partially, these genomic differences. *Osedax* and its endosymbiont co-depend on and compete to exploit the finite, nutritionally unbalanced diet obtained from bones, which might have favoured the evolution of an energetically "cheaper" genome in *Osedax*. In Vestimentifera, however, the endosymbiont acts as a primary producer, which might be able to sustain larger host genomes. Drawings are not to scale.

repertoire, with gene losses and expansions in key functional groups that support metabolic adaptations to its symbiotic lifestyle (Figs. 2, 3b, 4a; Supplementary Fig. 5d, f). As observed in the marine microbial assemblages on bone surfaces[48], the expansion of secreted matrix metalloproteases[33] (Fig. 5a) combined with the active secretion of acid in the root tissue[32] are the most probable mechanisms of bone digestion by the host (Fig. 7a). The *Osedax*-microbe association, however, entails further molecular and metabolic interactions to overcome a nutritionally unbalanced diet that is deficient in carbohydrates but enriched in (hydroxy)proline- and glycine-rich proteins and, in some cases, lipids[49,50]. Most notably, our findings suggest that the *Oceanospirillales* endosymbionts might be able to provide *Osedax* with glucose through the glyoxylate cycle (Fig. 3b) and that *Osedax* and the endosymbionts cooperate to maintain a physiological status of hyperammonemia (Fig. 4d). The former allows the catabolism of fatty acids to produce carbohydrates, which the host could take up by digesting the endosymbionts and store as glycogen (e.g., as seen in *Osedax*'s oocytes[74]), whereas the latter could stimulate the biosynthesis of amino acids, ultimately counterbalancing the lack of carbohydrates and skewed amino acid composition in bone. Notably, the use and occurrence of the glyoxylate cycle in animals is controversial[75,76] and only reported in a handful of taxa[77,78], likely as a consequence of horizontal gene transfer[79] and often concerning stress and a metabolic diapause, such as in the Dauer larva of nematodes[51], hibernating mammals[80] and bleached coral[81]. Indeed, *Osedax*, like Vestimentiferan hosts and their endosymbionts, lacks isocitrate lyase, but this enzyme is present in *Osedax*'s endosymbiont[18] (Fig. 3b; Supplementary Data 3, 7). Therefore, the

metabolic diversity of the *Oceanospirillales* endosymbiont may be critical to maximising the use of the imbalanced resources derived from the bones and ultimately acts as a selective pressure to acquire and maintain this microbe as the primary symbiont.

Symbiotic interactions can impose selective pressures that direct genome evolution—most notably in symbionts[82] but also occasionally in hosts[83]—triggering changes in genome size (e.g., genome erosion)[84], gene content[85] and even DNA base composition in favour of AT-rich genomes[86]. Most of these changes, however, are known for strictly vertically transmitted obligate endosymbionts of insects. Our study shows that Vestimentifera and *Osedax*, two annelid lineages within Siboglinidae that establish environmentally acquired symbioses, show differences in genome structure and composition (Fig. 2a–c; Supplementary Fig. 2m). While Vestimentifera tends to have larger genome sizes, similar GC content to asymbiotic annelids[37,38], and larger gene repertoires, *O. frankpressi* has a small, AT-rich genome, with a reduced gene content (Fig. 7b). In addition, these Siboglinidae crucially differ in their nutritional symbioses—chemoautotrophic in Vestimentifera and heterotrophic in *Osedax*—which enable them as adults to thrive in different ecological niches with different nutritional pressures. In hydrothermal vents and methane seeps, Vestimentifera relies on virtually unlimited inorganic nutrients that are exploited by the endosymbionts, which in their role as primary producers sustain long-lasting collaborative co-dependencies with their hosts[3,5]. Decaying bones are, however, nutritionally finite, and thus *Osedax* and their endosymbionts may establish a competitive co-dependency to exploit those nutritionally unbalanced resources (Fig. 7b). Moreover, the potential use of the glyoxylate cycle for energy production would be

less energetically efficient than the sole catabolism of fatty acids[87]. Therefore, we hypothesise that the interaction between *Osedax* and its endosymbiont might, in turn, favour the genomic streamlining of the annelid host (Fig. 7b) so that it becomes metabolically and energetically "cheaper" and can sustain larger endosymbiotic populations for longer periods. Our findings thus suggest that incipient genome erosion can occur in hosts with horizontally acquired symbionts and that adaptive genome evolution may differ based on the type of nutritional interactions between the host and symbiont. In the future, dissecting the metabolic co-dependencies between Siboglinidae and their endosymbionts, including the Frenulata and *Sclerolinum*—the other two major lineages within Sibogliniade—will help to disentangle the role of neutral and adaptive selective pressures in the evolution of these fascinating, but still poorly understood, animal symbioses.

## Methods

### Specimen collections, gDNA extraction and sequencing
Live adult specimens of *O. frankpressi*, *Oasisia alvinae* and *R. pachyptila* were obtained with deep-sea specialised robots off the coasts of California and Mexico (Supplementary Fig. 1c, d). Mexican samples were collected under CONAPESCA permit PPFE/DGOPA-200/18. Ultra-high molecular weight genomic DNA (gDNA) was extracted following the Bionano Genomics IrysPrep agar-based, animal tissue protocol (Catalogue # 80002) from an entire *O. frankpressi* adult female, a piece of the trunk (including trophosome) of *Oasisia alvinae*, and a piece of the vestimentum of *R. pachyptila*. Long-read PacBio sequencing and short-read Illumina sequencing was performed at the Genome Centre of the University of California Berkeley in a PacBio Sequel II and Illumina Novaseq platforms (Supplementary Table 1).

### Transcriptome sequencing
Total RNA from dissected tissues and body parts of *Oasisia alvinae* (crown, opisthosome and trophosome), and *R. pachyptila* (crown and trunk wall) was extracted with an NEB totalRNA Monarch kit and used for standard strand-specific RNA Illumina library prep. Libraries were sequenced to a depth of 40–50 million paired reads of 150 bases length in a NovaSeq platform (Supplementary Table 1). Publicly available datasets for *O. frankpressi* (NCBI short read archive accession numbers SRR2017399 and SRR2017400) were used in this study (Supplementary Table 1).

### Host genome assembly and quality check
PacBio reads were used to generate an initial genome assembly with Canu v.1.8[88] with options 'batOptions = "-dg 3 -db 3 -dr 1 -ca 500 -cp 50'. Two rounds of polishing using PacBio reads were performed using Pbmm2 v.1.1.0 (https://github.com/PacificBiosciences/pbmm2) and Arrow (pbgcpp v.1.9.0)[89]. Short genomic Illumina reads were quality filtered with FastQC v.0.11.8 and Cutadapt v.2.5[90], mapped to the polished assembly with BWA v.0.7.17[91] and used for final polishing with Pilon v.1.23[92]. The polished versions of the genomes of *O. frankpressi*, *Oasisia alvinae* and *R. pachyptila* were used as input to BlobTools v.2.1[93] to identify and remove contigs with high similarity to bacteria. After decontamination, the haplotypes were purged with Purge_Dups v.1.0.1[94]. Quality check was performed with BUSCO v.3.0.2[95], to estimate gene completeness of the assembly (Supplementary Table 3), QUAST v.5.0.2[96], and KAT v.2.4.2[97] to assess haplotype removal (Supplementary Fig. 2b–d) and potential bacterial remnants.

### Genome size estimations
Short Illumina reads were mapped to the reference host genome assembly with BWA v.0.7.17 and KAT v.2.4.2[97] to count and generate a histogram of canonical 21-mers. GenomeScope2[98] was used to estimate the genome size and heterozygosity (Supplementary Fig. 2e–g).

### Symbiont genome assembly and annotation
For *O. frankpressi* and *Oasisia alvinae*, we used Kraken2 v.2.1.0[99] and Krakentools v.0.1[99] to isolate long PacBio reads of bacterial origin. After error correction with Canu v.1.8[88], these PacBio reads were assembled using Metaflye v.2.9[100] followed by ten polishing iterations with options "–pacbio-corr –meta –keep-haplotypes –iterations 10" and final polishing with NextPolish v.1.4.0[101]. The resulting assemblies were manually inspected using Bandage v.0.9.0[102], binned with Max-Bin2 v.2.2.7[103] and quality checked with CheckM v.1.0.8[104] and Meta-Quast v.5.2.0[105]. Gene annotation was performed with Prokka v.1.14.5[106] with the "–compliant" option and proteins involved in secretion systems were identified by scanning for unordered replicons using the curated HMM profiles of TXSscan in MacSyFinder v.2[107]. The bacterial genomes were checked for secreted proteins with eukaryotic-like domains using EffectiveELD through EffectiveDB, on default settings[108]. All coding sequences of the main endosymbiont ribotype for *O. frankpressi*, Vestimentifera and Frenulata were assigned KO numbers using BlastKOALA v.2.2[109], which were used as input for KEGG Mapper v.5[110] to analyse the metabolic capabilities of each symbiont. The NCBI COG database[111] was used to tag functional categories to the annotated genes. Enrichment analyses of functional categories and Gene Ontology terms were performed with GSEA v.4.2.3[112] and OrthoVenn2 v.2[113]. To compute the *p*-values for enriched Gene Ontology terms in a protein cluster (Supplementary Data 6, 7), a hypergeometric distribution was used to identify significantly enriched terms within each cluster of orthologous/paralogous genes. GTDB-Tk v.1.6.0[114] was used for whole genome phylogenetic placement and identification of neighbouring available genomes isolated from free-living deep-sea bacteria. Circos v.0.69-9[115] was used for genome assembly visualisation.

### Annotation of repeats in host genomes
RepeatModeler v.2.0.1[116] and Repbase[117] were used to build a de novo library of repeats for the host genome of *O. frankpressi*, *Oasisia alvinae* and *R. pachyptila*. The predicted genes of *Owenia fusiformis*[37] and DIAMOND v.0.8.22[118] were used to filter out bona fide genes in the predicted repeats with an e-value threshold of 1e-10. Subsequently, RepeatMasker v.4.1.0[119] (Supplementary Tables 4–6) and LTR-finder v.1.07[120] were used to identify and annotate repeats, and RepeatCraft[121] to generate a consensus annotation that was used to soft-mask the genome assemblies of the three annelid species. To explore the transposable element landscape, we used the online tool TEclass[122] to annotate the TEs identified by RepeatModeler and the scripts "calc-DivergenceFromAlign.pl" and a custom-modified version of "create-RepeatLandscape.pl", both from RepeatMasker v.4.1.0, to estimate Kimura substitution levels, which were plotted using ggplot2 v.3.3.0[123]. Previously published TE landscapes were included for comparisons[37].

### Functional annotation of host genomes
Individual RNA-seq Illumina libraries (Supplementary Table 1) were de novo assembled with Trinity v.2.9.1[124] after quality trimming with Trimmomatic v.0.35[125]. GMAP v.2017.09.30[126] and STAR v.2.7.5a[127] were used to map transcripts and quality-filtered Illumina reads to the soft-masked genome assemblies of the corresponding species. For *R. pachyptila*, publicly available datasets (SRA accession numbers SRR8949056 to SRR8949077) were also mapped to the soft-masked genome assembly. In addition, gene transfer format (GTF) files from the mapped reads and curated intron junctions were inferred with StringTie v.2.1.2[128] and Portcullis v.1.2.2[129]. All RNA-seq-based gene evidence was merged with Mikado v.2.Orc2[130], which produced a curated transcriptome-based genome annotation. Full-length Mikado transcripts were used to train Augustus v.3.3.3[131], which was then used to generate ab initio gene predictions that incorporate the intron hints of Portcullis and the exon hints of Mikado. In addition, Exonerate v.2.4.0[132] was used to produce spliced alignments of the curated

proteomes of *Owenia fusiformis*, *C. teleta* and *L. luymesi* that were used as further exon hints for Augustus. Finally, the Mikado RNA-seq-based gene evidence and the ab initio predicted Augustus gene models were merged with PASA v.2.4.1[133]. A final, curated gene set was obtained after removing spurious gene models and genes with high similarity to transposable elements. Gene completeness and annotation quality were assessed with BUSCO v.3.0.2[95]. Trinotate v.3.2.1[134], PANTHER v.1.0.10[47] and the online tool KAAS[135] were used to functionally annotate the curated gene sets.

## Comparison of *R. pachyptila* assemblies

Overall genomic stats were obtained with BUSCO v.3.0.2[95], QUAST v.5.0.2[96] and AGAT v.0.5.0[136]. We used minimap2 v.2.17 to align our *R. pachyptila* assembly with the assembly previously reported[10] and the R package pafr to generate a dot-plot representation of the sequence similarity between the two versions. In addition, we reassembled all transcriptomic evidence published elsewhere[12] using Trinity v.2.9.1[124] and cd-hit v.4.8.1[137]. To identify one-to-one orthologs between genomic and transcriptomic resources, we used a reciprocal best BLAST hit approach with BLAST v.2.12.0+ [138]. Finally, we used PFAMscan v.1.6[139] to identify and quantify distinct Pfam domains in the different assemblies.

## Gene family evolutionary analyses

The non-redundant proteomes of *O. frankpressi*, *Oasisia alvinae* and *R. pachyptila* together with 25 high-quality genomes spanning major groups of the animal tree (Supplementary Data 2) were used to construct orthogroups with OrthoFinder v.2.5.2[140] using DIAMOND v.2.0.9[118] with "–ultra-sensitive" option. The OrthoFinder output and a published Python script[36] were used to infer gene family evolutionary dynamics at each node and tip of the tree. Gene Ontology term enrichment analyses for expanded and lost gene families were performed with the R package "TopGO" v.2.42.0. The number of orthologs per gene family and species as generated by OrthoFinder was used to perform a Principal Component Analysis with R built-in functions.

## Reconstruction of host metabolic pathways and developmental gene sets

PANTHER and Pfam annotations obtained through PANTHER v.1.0.10[47] and Trinotate v.3.2.1[134], respectively, were used to assess for the presence of each enzyme involved in the synthesis of amino acids, vitamin Bs, nitrogen metabolism, glycine degradation, matrix metalloproteases, transcription factors and DNA repair pathways in an array of annelid species. A combination of BlastKOALA[109] and KofamKOALA[141] was used to annotate the host and endosymbiont genomes for the analysis of the lipid and carbohydrate metabolism. Information about each step in a pathway was collected from MetaCyc[142], KEGG[143] and PANTHER[47] databases. To analyse the tissue-specific expression of candidate genes in *O. frankpressi*, *Oasisia alvinae* and *R. pachyptila*, quality-filtered short Illumina reads were pseudo-mapped to the filtered gene models of each species with Kallisto v.0.46.2[144] to quantify transcript abundances as Transcripts per Kilobase Million (TPM) values. The R libraries ggplot2 v.3.3.0[123] and pheatmap v.1.0.12 (https://cran.r-project.org/web/packages/pheatmap/index.html) were used to plot expression and abundance heatmaps.

## Reconstruction of innate immune repertoires

The OrthoFinder output was used to identify gene families of innate immune pattern recognition receptors of *O. frankpressi*, Vestimetifera and two asymbiotic annelids, *Owenia fusiformis* and *C. teleta*, with the published pattern recognition receptors of Vestimetifera[9] as baits (Supplementary Data 11–17). PANTHER and Pfam annotations (see above) of the target proteins were further used to remove sequences that were too short or lacked target domains. TPM expression values

(see above) and TBtools v.1.042[145] were used to plot gene expression heatmaps.

## Reconstruction of the G protein-coupled receptor (GPCR) repertoire

Transcriptomes of the focal species were downloaded and processed as described elsewhere[146]. Multiple sequence alignments of rhodopsin type GPCRs (PF00001), secretin type GPCRs (PF00002), glutamate type GPCRs (PF00003) and frizzled type GPCRs (PF01534) were downloaded from the Pfam webpage (https://pfam.xfam.org) and used to create HMM profiles using hmmer-3.1b2[147]. HMMer search was performed with an e-value cut-off of 1e-10. The online version of CLANS (https://toolkit.tuebingen.mpg.de/tools/clans) was used for the initial BLAST comparison for the cluster analysis and edges below 1e-10 for secretin, glutamate and frizzled type GPCRs and 1e-20 for rhodopsin type GPCRs were removed. The java offline version of CLANS[148] was then used for the cluster analysis. The *p*-value for clustering was set to 1e-25. Singletons and group-specific sequence clusters with less than five sequences and no annotation (using Linkage clustering for identification) were removed. The highly vertebrate-specific expanded olfactory GPCR type-A receptors were also deleted as these showed no connections and strongly repulsed all other sequences. Gene clusters were annotated according to the presence of characterized sequences of *Drosophila melanogaster*, *Homo sapiens*, *Danio rerio* and *Platynereis dumerilii*.

## Orthology assignments

MAFFT[149] with default options was used to align candidate sequences to a curated set of proteins that we obtained either from previous studies[36,150] or manually from UniProt[151]. Conserved protein domains were retained by trimming by hand the alignment in Jalview[152] and the resulting sequences were re-aligned in MAFFT with the "L-INS-I" algorithm[149]. After a final trim to further remove spurious regions with trimAI v.1.4.rev15[153], FastTree v.2.1.10[154] with default options and IQ-Tree v.2.2.0-beta[155] (for matrix metalloproteases) using the options "-m MFP -B 1000", were used to infer orthology relationships. In addition, for the matrix metalloproteases, the posterior probabilities were obtained from Bayesian reconstructions in MrBayes v.3.2.7a[156], which were performed using as a prior the LG matrix[157] with a gamma model[158] with four categories to describe sites' evolution rate. Four runs with eight chains were run for 20,000,000 generations. FigTree v.1.4.4 (https://github.com/rambaut/figtree) and Adobe Illustrator were used to edit the final trees. CD-Search[159] with default options and the Conserved Domain Database (CDD)[160] were used to annotate protein domains in the predicted matrix metalloproteases.

## Reporting summary

Further information on research design is available in the Nature Portfolio Reporting Summary linked to this article.

# Data availability

All sequence data associated with this project are available at the European Nucleotide Archive (project PRJEB55047). This study also used previously published datasets with accessions SRR2017399, SRR2017400, SRR8949056–SRR8949077 [https://www.ncbi.nlm.nih.gov/bioproject/PRJNA534438]. Additional files are publicly available at https://github.com/ChemaMD/OsedaxGenome. Source data are provided as a Source Data file. Source data are provided with this paper.

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

## Acknowledgements

We thank members of the Martín-Durán and Henry lab for support and discussions, as well as Gustavo A. Ballén, Ferdinand Marlétaz and the core technical staff at the Department of Biology at Queen Mary University of London for their support. This research utilised Queen Mary's Apocrita HPC facility, supported by QMUL Research-IT (https://doi.org/10.5281/zenodo.438045). Many thanks to Chief Scientists Victoria Orphan and Bob Vrijenhoek, the captains and crews of the R/V *Western Flyer* and R/V *Falkor* and the pilots of the ROVs *Tiburon* and *SuBastian* for crucial assistance in specimen collection. Collections for this project were enabled by the Monterey Bay Aquarium and Research Institute and the Schmidt Ocean Institute. This work was funded by a Wellcome Trust Seed Award in Science to JMM-D (213981/Z/18/Z) and a NERC IRF awarded to LMH (NE/M018016/1). JWQ, PYQ, and YNS were supported by the Key Special Project for Introduced Talents Team of Southern Marine Science and Engineering Guangdong Laboratory (Guangzhou) (GML2019ZD0409) and the Major Project of Basic and Applied Basic Research of Guangdong Province (2019B030302004). AMC was funded by a Scripps Postdoctoral Fellowship.

## Author contributions

J.M.M.-D., L.H., G.R., and G.M. conceived and designed the study. G.R., N.R.-K. and S.G. collected the samples; G.M. assembled and annotated all genomes, and performed gene family evolution and metabolic complementarity analyses; B.P. assembled and annotated the symbiont genomes and contributed to metabolic complementarity analyses; Y.S., P.Q. and J.Q. performed analyses on PRR evolution; D.T. and G.J. did GPCR evolutionary analyses; F.M.M.-Z. performed Bayesian phylogenetic analyses; M.T. performed genomic extractions; A.M.C. and Martin Tresguerres contributed to host-symbiont metabolic analyses; G.M., L.H., and J.M.M.-D. drafted the manuscript and all authors critically read and commented on the manuscript.

## Competing interests

The authors declare no competing interests.
