## [Peer Review File · Nature Communications]

Distinct genomic routes underlie transitions to specialised symbiotic lifestyles in deep-sea annelid wormsREVIEWER COMMENTS

Reviewer #1 (Remarks to the Author):

In this study, Martín-Durán and coworkers conducted a comprehensive comparative analysis of the physiology of different marine annelids and their endosymbionts to better understand particular metabolic interactions and adaptations of the symbiont partners. Such a systematic comparative genomic approach is interesting and could contribute to determine new genetic and molecular principles of symbiotic lifestyles. The data of this work indicate that the endosymbionts of *Osedax frankpressi* can compensate putative metabolic deficiencies of the host. It was also found that, unlike in chemoautotrophic symbioses, innate immunity genes to acquire and control endosymbionts are reduced in *O. frankpressi*. Finally, an expansion of matrix metalloproteases to digest collagen was determined.

Thus, this comprehensive study is characterized by a detailed genomic analysis of the *Osedax frankpressi* holobiont and related annelid genomes. It is worth mentioning that this diligent work resulted in numerous complex and beautiful figures.

My major concern is the comparison of the two totally different metabolic strategies of the chemolithoautotrophic *Riftia pachyptila* and the bone-eating, heterotrophic *Osedax frankpressi* symbioses in this study. The gain of knowledge of this approach is in my opinion limited. Most of the stressed metabolic differences are obvious and expected.

I think the study would benefit from focusing on the putative metabolic and molecular interactions between *Osedax frankpressi* and its bacterial microbiome.

The authors' main hypothesis of host genome evolution based on the observed reduced innate immune response is not convincing. One could also assume that an enhanced immune response in the Vestimentifera indicates greater containment of the symbiont population by the host, or efforts to maintain symbiotic equilibrium.

Further comments:

The authors cite three studies of sequenced Vestimentiferan genomes from hydrothermal vents and mention that these studies complement their genome sequencing efforts. It is worth mentioning that there also exists a host transcriptome sequencing project of *Riftia pachyptila* which has been recently published (Hinzke et al. mBIO 2019) and which is also cited by the authors later in the manuscript, where different tissues of this tubeworm were investigated. In this respect it would be useful to see how the *Riftia* transcriptome analysis of this study compares to the *Riftia pachyptila* transcriptome analysis of Hinzke et al. (mBIO 2019).

Figure 3: The authors present in this figure a comparison of functional protein groups from the endosymbionts of different annelids. As it is mentioned in the manuscript that different bacterial symbionts are involved in the *O. frankpressi* and *Oasisia alvinae* symbioses. Due to this fact, it should be discussed whether this direct functional comparison with the monoclonal chemolithoautotrophic symbiont populations of, e.g., *R. pachyptila* is compelling.

Page 9, lines 202-203: It is stressed that "A comparison of *Osedax*'s endosymbiont to those of Vestimentifera reveals significant differences in their metabolic capabilities." This might be not so surprising since you compare heterotrophic and chemolithoautotrophic bacterial symbionts. Thus, the observed higher number of genes involved in the metabolism and transport of amino acids, coenzymes, lipids, and carbohydrates in the *Osedax* symbiont makes sense.

Page 9, lines 223-225: The authors claim that "the heterotrophic endosymbiont of *Osedax* has a much more versatile metabolism than the chemolithoautotrophic bacteria of Vestimentifera." I disagree, please keep in mind that also the endosymbiont of *R. pachyptila* might be able to live partially heterotrophically. Considering the chemolithotrophic pathways of the *Riftia* symbiont I assume that the metabolic capacities of both symbionts might be in sum rather similar although functionally different, of course.

Figure 6: Please specify in the legend the source of the genome data used in this analysis. It looks

like, except for *O. frankpressi* and *R. pachyptila*, all other sequence data were produced by other groups. If this is the case, please add the appropriate references in the legend of this figure.

Figure 7 is beautiful, but difficult to read.

Please check reference "73 Andrews, S. (2010)." and "121 Bioconductor — TopGO (2022).".

Supplementary Table 1. Please add the individual accession numbers of the listed genomic and transcriptomic data sets. Are all sequence data submitted to a public data base?

Supplementary Table 2. Please add in the table the accession numbers of the listed microbial genomes sequenced from *O. frankpressi*.

Reviewer #2 (Remarks to the Author):

This is an excellent and beautifully illustrated paper that provides insight into the functionality of symbiont-bearing deep-sea annelid worms, based on genomes and transcriptomes of the bone-eating *Osedax* and two species of vestimentiferan tubeworms. The authors discuss the genetic changes that lead to distinct symbiotic lifestyles, highlighting gene family gains and losses that shape the evolution of siboglinids, such as the expansion of immune genes in vestimentiferans, and unique genomic adaptations for bone digestion, such as an array of matrix metalloproteases in *Osedax*. They underline the interactions and co-dependencies of the hosts and their symbionts and explore the developmental toolkit in these worms. These results lead to a beautiful hypothesis, suggesting that competitive co-dependency between the host and its endosymbiont may lead to genome streamlining in the former. I enjoyed reading this paper, describing fascinating life forms, whose biology and evolution are exciting not only to the specialists but also to the broad readership. I recommend its acceptance.

Two minor issues:

L77 – "Untangling"?

L137 – "Therefore, *O. frankpressi* has the smallest genome of all sequenced Vestimentifera." should be "Therefore, *O. frankpressi* has the smallest genome of all sequenced siboglinids."

Maxim Rubin-Blum

Reviewer #3 (Remarks to the Author):

The manuscript 'Distinct genomic routes underlie transitions to specialised symbiotic lifestyles in deep sea annelid worms' by Martín-Durán and co-authors investigates and compares the genomes of the deep-sea whale fall and hydrothermal vent siboglinids. Further, the authors study the genome of their heterotrophic and chemotrophic symbionts and compare them to other bacteria to understand genomic features that support a symbiotic bacterial life style.

Concerning the animals, the genome of *Osedax* and *Oasisia* would be the first to be published, the genome of *Riftia* has been very recently published already (de Oliveira et al. 2022). This previous paper not only is better in terms of contiguity – N50, number of scaffolds than the presented *Riftia* genome in this manuscript, at large both the presented results confirm the already published ones. These facts are not properly cited, neither in the introduction where the reader would expect to get an overview of what is known already nor is it then incorporated thoroughly in the results and discussion part, especially not for major findings like the developmental gene screenings. In the figure 8, everything the authors described in *Riftia* has been previously published. It is literally the same result. Even the identification of enzymes related to amino acid biosynthesis is agreeing completely with previous paper, but not cited. This clearly gives a false impression.

In addition, the genomes of long-lived, slow growing *Lamellibrachia luymesii* (Li et al. 2019) and

Paraescarpia echinospica (Sun et al. 2021) are from vestimentiferans from hydrocarbon seeps and therefore are from very different habitats than hydrothermal vents with short-living, fast growing Riftia. While I can understand that it is difficult to incorporate a new published paper when all analyses are already performed and the text in progress of writing, here clearly the path of downplaying was chosen and this is not correct. To simply state in the introduction that 'our understanding of the genetic traits that sustain symbioses in Siboglinidae is scarce and currently limited to Vestimentifera' is not enough to acknowledge three very different and detailed genome publications.

Overall, the bioinformatics part of the paper is really good. There are no major flaws, the pipeline follows a logical path and includes all the required steps for pre- and post-processing of PacBio and Illumina data.

In the lines 320-323, the authors insisted twice that immune-related genes are expanded in Vestimentifera, however, that is not the case in either the draft Riftia genome available in this paper, neither in previous publication. I suggest to rephrase the sentence to better describe the reality of the findings. The same applies for the conclusion paragraph (Line 329 - 331). The authors are not consistent throughout the paper.

The authors state that *O. frankpressi*, in contrast to vestimentiferans and other annelids, show gene losses and expansions in key functional groups that support metabolic adaptations to its symbiotic lifestyles. This, however, is the case for every vestimentifera paper in the literature. Lamellibrachia, Paraescarpia, Riftia, all described gene families that are contracted and other that are expanded and linked this to the symbiosis and habitat. The authors also state that Vestimentifera experienced an event of gene family expansion in its last common ancestor (which is true and showed in the Supp. Figure 4B). However, this is not the same to say that Riftia, Paraescarpia and Oasisia STILL have an expanded repertoire of x,y and z. Their own Supp. Figure 4B shows that Riftia, Paraescarpia and Oasisia are losing way more genes than gaining and thus experiencing maybe reductive evolution in their genomes. Overall, all this points to the authors' are trying too hard to show that Osedax is incredibly different from other systems, but in fact this has not been shown clearly. Instead, there might be more similarities than differences. The authors might consider to discuss this topic more openly.

Concerning the symbionts – Also there is no overview provided in the introduction. Further, the authors state that 'we sequenced the hologenome of *Osedax frankpressi* Rouse, Goffredi & Vrijenhoek, 2004, as well as that of two vent dwelling Vestimentifera, *Oasisia alvinae* Jones, 1985 and *Riftia pachyptila* Jones, 1981'. They say that the "hologenome" is sequenced, but they don't cite where it has been published and by who. Also, here I would expect the citation of all genomes of endosymbionts including the most recent one of the Riftia from de Oliveira et al. 2022. While the primary symbiont of *Osedax* is presented quite extensively, that of *Oasisia* is not. Also, I miss the annotation of *Oasisia* and *Riftia* symbiont in like Supplementary Table 12 done for *Osedax* symbiont.

The authors use the word chemoautotrophic and chemotrophic as a synonym, not considering that a chemotroph can also be a heterotroph (as has been shown for a ciliate symbiont), hence a chemoheterotroph. Chemotrophs can be autotrophs, heterotrophs or mixotrophs. Fact is that *Endoriftia* shows many genes that support autotrophy (their function well supported since the very beginning of this research) as well as heterotrophy. The functioning as heterotroph symbiont has not been shown clearly yet, but since this is a genome paper the presence of genes for heterotrophy should be cited. I would really question the conclusion that the heterotrophic endosymbiont of *Osedax* is more metabolic versatile than *Endoriftia* and other vestimentiferan endosymbionts.

As a more theoretical point – while the use of the term hologenome might be justified for symbioses with vertically transmitted, obligate symbionts without free-living populations, because selection acts on the holobiont only, this is clearly not the case for symbioses with horizontally transmitted symbionts – with host-associated and free-living populations and host stages that are symbiotic as well as without their symbiont during early development. Here, selection acts on multi-levels (as in siboglinids) and therefore the term hologenome is rejected by many researchers.

Due to all the reasons outlined above I suggest a major revision.

Reviewer #4 (Remarks to the Author):

There was a lot of great work that went into this manuscript and it is a good look at the genomes of these animals. However there is one glaring omission. The genome of Lamellibrachia and the genomes of frenulate endosymbionts were not include or really even discussed in any of the analyses. This is a bit shocking as the authors are very much aware of this work and, more importantly the comparison between Osedax and Lamellibrachia is a better comparison as the water chemistry and environment of a whale fall and a seep are more similar than a whale fall and vents. This is a major oversight and should be corrected prior to publication. Granted the Lamellibrachia genome is a Illumina and not a PacBio genome but is it still good enough for many of the comparisons. Why the Li et al (2018) was excluded is not clear. Both of these papers (below) are very germane to the discussions in this paper.

Also this paper suffers from the genomic shotgun approach meaning the manuscript tried to report on numerous genomic features that just are not relevant to the main story (ie Hox gene discussion, or why 28 genomes across metazoa are included). I will spare the other examples.

The flow of the paper could be improved and several sentences are written awkwardly - especially in the Abstract. For example line 34 starts "endosymbionts ultimately...." and the reader does not know which endosymbionts are being discussed until the very end of the sentence. Or lines 126 and 129 which are very confusing on the repeat content of Capitella.

Line 144 it mentions that a PCA of genomes was conducted -- it is not at all clear what the input data was for the PCA -- what is being compared?

The main text needs more on the quality of the genomes - N50, where they dovetailed or optically mapped? The low BUSCO for Osedax is concerning. In my experience with PacBio that signals a poor assembly. Yes they did find them by manually but this is a point of concern about quality.

Li Y, Tassia MG, Waits DS, Bogantes VE, David KT, Halanych KM. 2019. Genomic adaptations to chemosymbiosis in the deep-sea seepdwelling tubeworm *Lamellibrachia luymesii*. *BMC Biol.* 17(1):91.

Li, Y., M. R. Liles, K. M. Halanych. 2018. Endosymbiont genomes yield clues of tube worm success. *ISME Journal.* 12:2785-2795. /doi.org/10.1038/s41396-018-0220-z

RESPONSE TO REVIEWERS' COMMENTS

We thank the four reviewers for their positive appraisals and constructive comments, which have significantly improved our work. We are pleased to provide a revised manuscript that we hope addresses the comments raised. The main changes to our manuscript are summarised as follows:

As recommended (reviewer #1), we have expanded our analyses on the putative molecular interactions between *Osedax* and its endosymbiont to explore all major metabolic pathways and symbiont secretion systems. Combining highly sensitive hidden Markov model (HMM) searches with genome-wide KEGG annotations, we have discovered that *Osedax*'s endosymbionts, but not the host, possess the glyoxylate cycle. This metabolic route is unusual in animals and allows the synthesis of carbohydrates (that are rare in bone) from the catabolism of fatty acids (that can be very abundant in bone and may act as energy storage in the host's roots). Remarkably, this potential metabolic interaction between host and symbiont cannot occur in Vestimentifera and Frenulata because all sequenced endosymbiont genomes lack at least one enzyme of the cycle. Therefore, these novel findings (now in main Figures 3 and 7) reveal an *Osedax*-specific metabolic adaptation that may contribute to overcoming a diet limitation and act as a potential mechanism for endosymbiont selection.

We have compared our assembly and annotation of the genome of *Riftia pachyptila* with the one reported by de Oliveira *et al.* 2022 and the reassembly of all the transcriptomic resources reported by Hinzke *et al.* 2019 (reviewers #1 and #3) and acknowledge more explicitly how our findings on *R. pachyptila* agree with previous studies (reviewer #3). In addition, we now include the genomic resources for *Lamellibrachia luyesi* and a frenulate endosymbiont genome in all our analyses (reviewer #4), further solidifying our previous conclusions.

Finally, we have extensively rewritten our manuscript to focus more on the molecular and metabolic interactions between *Osedax* and its endosymbiont, review previous genomic knowledge on Siboglinidae and shorten the sections about the developmental toolkit and DNA repair pathways accordingly (reviewers #1, #3 and #4). We have added a complete annotation of *Oasisia alvinae*'s endosymbiont (reviewer #3) and qualified our interpretations on the reduced immune complement of *Osedax* (reviewer #1) and the metabolic versatility of its endosymbiont (reviewers #1 and #3). We have also incorporated all other text and figure suggestions and formatted the manuscript to Nature Communications' requirements. All main findings are summarised in 7 main display items, 16 supplementary figures and 27 supplementary tables.

Below we provide a detailed point-by-point response to each reviewer's concerns. We hope we have addressed them all satisfactorily and will happily address any additional comments and suggestions.

Point-by-point response

Reviewer #1 (Remarks to the Author):

*In this study, Martín-Durán and coworkers conducted a comprehensive comparative analysis of the physiology of different marine annelids and their endosymbionts to better understand particular metabolic interactions and adaptations of the symbiont partners. Such a systematic comparative genomic approach is interesting and could contribute to determine new genetic and molecular principles of symbiotic lifestyles. The data of this work indicate that the endosymbionts of *Osedax frankpressi* can compensate putative metabolic deficiencies of the host. It was also found that, unlike in chemoautotrophic symbioses, innate immunity*

genes to acquire and control endosymbionts are reduced in O. frankpressi. Finally, an expansion of matrix metalloproteases to digest collagen was determined.

Thus, this comprehensive study is characterized by a detailed genomic analysis of the Osedax frankpressi holobiont and related annelid genomes. It is worth mentioning that this diligent work resulted in numerous complex and beautiful figures.

RESPONSE: Many thanks for the positive appraisal.

My major concern is the comparison of the two totally different metabolic strategies of the chemolithoautotrophic Riftia pachyptila and the bone-eating, heterotrophic Osedax frankpressi symbioses in this study. The gain of knowledge of this approach is in my opinion limited. Most of the stressed metabolic differences are obvious and expected. I think the study would benefit from focusing on the putative metabolic and molecular interactions between Osedax frankpressi and its bacterial microbiome.

RESPONSE: Following the reviewer's advice, we have expanded our analyses to identify potential molecular interactions between *Osedax* and its endosymbionts at a genome-wide scale. This has allowed us to consider metabolic pathways other than those involved in amino acid biosynthesis, nitrogen recycling and vitamin production and identify complementary traits in their gene repertoires that might have a potential selective advantage. By doing so, we have discovered, among other things, that *Osedax*'s endosymbionts possess the entire glyoxylate cycle, which could enable these two organisms to synthesise carbohydrates from fatty acids. We hypothesise that this might be an essential adaptation that compensates for the carbohydrate deficiency of bones and may act as a mechanism to select the endosymbiont. These findings are now in the new main Figure 3 and lines 255–278.

We have also applied this approach to the four genomes of Vestimentifera (host and symbiont). As the reviewer correctly points out, Vestimentifera and *Osedax* have different metabolic strategies but are closely phylogenetically related. Therefore, its comparison is essential to identify which traits and interactions are ancestral to Siboglinidae (potentially involved in maintaining a bacterial endosymbiont) and which are lineage-specific and might thus represent nutritional adaptations. Despite its limitations, this comparison demonstrates that the presence of the glyoxylate cycle and the expansion of matrix metalloproteases occurred solely in the symbiosis of *Osedax*, reinforcing that they might represent essential adaptations to its unique bone-eating lifestyle.

The authors' main hypothesis of host genome evolution based on the observed reduced innate immune response is not convincing. One could also assume that an enhanced immune response in the Vestimentifera indicates greater containment of the symbiont population by the host, or efforts to maintain symbiotic equilibrium.

RESPONSE: We have amended our interpretation of *Osedax*'s reduced immunity complement (lines 371–375).

Further comments:

The authors cite three studies of sequenced Vestimentiferan genomes from hydrothermal vents and mention that these studies complement their genome sequencing efforts. It is worth mentioning that there also exists a host transcriptome sequencing project of Riftia pachyptila which has been recently published (Hinzke et al. mBIO 2019) and which is also cited by the authors later in the manuscript, were different tissues of this tubeworm were investigated.

In this respect it would be useful to see how the Riftia transcriptome analysis of this study compares to the Riftia pachyptila transcriptome analysis of Hinzke et al. (mBIO 2019).

RESPONSE: We thank the reviewer for pointing out that this was unclear. All transcriptomic datasets generated in Hinzke *et al* 2019 were included during the genome annotation step of our *R. pachyptila* assembly (lines 577–579: “For *R. pachyptila*, publicly available datasets (SRA accession numbers SRR8949056 to SRR8949077) were also mapped to the soft-masked genome assembly.”). In addition, we have made a new Supplementary Figure 3 comparing our assembly and annotation to the previously published in de Oliveira *et al.* 2022. We also include a comparison to a new assembly of all transcriptomic evidence by Hinzke *et al.* 2019, which demonstrates that 41.77% of the genes we annotated (15,469) have a direct counterpart in the transcriptomic data, despite the intrinsic redundancy and fragmentation of transcriptomic datasets (i.e., the comparison between the two genome assemblies yields a much higher similarity). Nevertheless, and as pointed out by reviewer #3, our findings on *R. pachyptila* broadly agree with and confirm the observations of de Oliveira *et al.* 2022 and Hinzke *et al.* 2019. We now state this explicitly in the manuscript (e.g., lines 143–146; line 282; lines 436–440).

Figure 3: The authors present in this figure a comparison of functional protein groups from the endosymbionts of different annelids. Is it mentioned in the manuscript that different bacterial symbionts are involved in the O. frankpressi and Oasisia alvinae symbioses. Due to this fact, it should be discussed whether this direct functional comparison with the monoclonal chemolithoautotrophic symbiont populations of, e.g., R. pachyptila is compelling.

RESPONSE: We now clarify this point. Although we recover multiple bacterial genomes from *O. frankpressi*, only one is an endosymbiont. The others are likely epibionts located on the epidermis of the trunk and crown (as we demonstrate in DOI: <https://doi.org/10.1101/2022.11.14.516544>) or part of the environmental microbiome associated with the bone. Therefore, all comparisons between endosymbionts consider only the monoclonal symbiont population (lines 549–550). The plots originally in Figure 3 are now in Supplementary Figure 7.

Page 9, lines 202-203: It is stressed that “A comparison of Osedax’s endosymbiont to those of Vestimentifera reveals significant differences in their metabolic capabilities.” This might be not so surprising since you compare heterotrophic and chemolithoautotrophic bacterial symbionts. Thus, the observed higher number of genes involved in the metabolism and transport of amino acids, coenzymes, lipids, and carbohydrates in the Osedax symbiont makes sense.

RESPONSE: We have now removed this statement and refocused our discussion in this section on our new results demonstrating *Osedax*’s endosymbionts contain unique metabolic properties that complement the host’s metabolism and that are not in the Vestimentifera symbioses. We also expanded our analysis of the symbiont secretion systems and revealed that all endosymbionts contain numerous protein secretion genes with eukaryote-like protein domains, many of which are lineage-specific (lines 218–224 and Supplementary Table 14). The graphs with the differences in the gene numbers involved in metabolism and transport (previously Figure 3b) have been moved to the supplementary material (Supplementary Figure 7).

Page 9, lines 223-225: The authors claim that “the heterotrophic endosymbiont of Osedax has a much more versatile metabolism than the chemolithoautotrophic bacteria of Vestimentifera.” I disagree, please keep in mind that also the endosymbiont of R. pachyptila might be able to live partially heterotrophically. Considering the chemolithotrophic pathways of the Riftia symbiont I assume that the metabolic capacities of both symbionts might be in sum rather similar although functionally different, of course.

RESPONSE: The reviewer is right. We have removed this statement and refocused our discussion on the unique metabolic traits of *Osedax* and its endosymbionts and how these differ from Vestimentifera, to understand these symbioses better.

Figure 6: Please specify in the legend the source of the genome data used in this analysis. It looks like, except for O. frankpressi and R. pachyptila, all other sequence data were produced by other groups. If this is the case, please add the appropriate references in the legend of this figure.

RESPONSE: Corrected as suggested (lines 1187–1188).

Figure 7 is beautiful, but difficult to read.

RESPONSE: To focus the text (reviewer #4), we have moved this figure to Supplementary Information. We have improved the new legend to clarify what each plot represents (“Each sequence is represented by a symbol and colour according to its species of origin. Grey lines indicate similarity between these sequences based on all-to-all BLAST searches, with the intensity of grey showing the Expect value (e-value) of the search.”).

Please check reference “73 Andrews, S. (2010).” and “121 Bioconductor — TopGO (2022).”.

RESPONSE: Thanks for pointing this out. These were computer programmes without published peer-reviewed journal articles. We have removed the references.

Supplementary Table 1. Please add the individual accession numbers of the listed genomic and transcriptomic data sets. Are all sequence data submitted to a public data base?

RESPONSE: All datasets have now been uploaded to the European Nucleotide Archive under the project with accession number PRJEB55047.

Supplementary Table 2. Please add in the table the accession numbers of the listed microbial genomes sequenced from O. frankpressi.

RESPONSE: We now indicate that all microbial genome assemblies in Supplementary Table 2 are in the European Nucleotide Archive in the project with accession number PRJNA813420.

Reviewer #2 (Remarks to the Author):

*This is an excellent and beautifully illustrated paper that provides insight into the functionality of symbiont-bearing deep-sea annelid worms, based on genomes and transcriptomes of the bone-eating *Osedax* and two species of vestimentiferan tubeworms. The authors discuss the genetic changes that lead to distinct symbiotic lifestyles, highlighting gene family gains and losses that shape the evolution of siboglinids, such as the expansion of immune genes in vestimentiferans, and unique genomic adaptations for bone digestion, such as an array of matrix metalloproteases in *Osedax*. They underline the interactions and co-dependencies of the hosts and their symbionts and explore the developmental toolkit in these worms. These results lead to a beautiful hypothesis, suggesting that competitive co-dependency between the host and its endosymbiont may lead to genome streamlining in the former. I enjoyed reading this paper, describing fascinating life forms, whose biology and evolution are exciting not only to the specialists but also to the broad readership. I recommend its acceptance.*

RESPONSE: Many thanks for the positive appraisal.

Two minor issues:

L77 – “Untangling”?

L137 – “Therefore, *O. frankpressi* has the smallest genome of all sequenced Vestimentifera.” should be “Therefore, *O. frankpressi* has the smallest genome of all sequenced siboglinids.”

Maxim Rubin-Blum

RESPONSE: We have changed these points as suggested.

Reviewer #3 (Remarks to the Author):

The manuscript ‘Distinct genomic routes underlie transitions to specialised symbiotic lifestyles in deep sea annelid worms’ by Martín-Durán and co-authors investigates and compares the genomes of the deep-sea whale fall and hydrothermal vent siboglinids. Further, the authors study the genome of their heterotrophic and chemotrophic symbionts and compare them to other bacteria to understand genomic features that support a symbiotic bacterial life style.

*Concerning the animals, the genome of *Osedax* and *Oasisia* would be the first to be published, the genome of *Riftia* has been very recently published already (de Oliveira et al. 2022). This previous paper not only is better in terms of contiguity – N50, number of scaffolds than the presented *Riftia* genome in this manuscript, at large both the presented results confirm the already published ones. These facts are not properly cited, neither in the introduction where the reader would expect to get an overview of what is known already nor is it then incorporated thoroughly in the results and discussion part, especially not for major findings like the developmental gene screenings. In the figure 8, everything the authors described in *Riftia* has been previously published. It is literally the same result. Even the identification of enzymes related to amino acid biosynthesis is agreeing completely with previous paper, but not cited. This clearly gives a false impression.*

RESPONSE: We now clarify that many of the results we obtained for *R. pachyptila* are consistent with and reproduce what has been previously shown in de Oliveira et al. 2022 (e.g., line 282; lines 436–440). In addition, we have expanded the introduction to reflect better what was previously known of the genomics of Vestimentifera (see the next point) and added additional references (lines 50–63). Following reviewer #1’s suggestion, we also include a new Supplementary Figure 3 comparing the two existing assemblies and the one large transcriptomic study for *R. pachyptila*. These analyses confirm that despite the assembly being less contiguous, our version is largely equivalent to the assembly of de Oliveira et al. 2022 (e.g., ~70% of the genes in de Oliveira et al. 2022 have an exact counterpart in our assembly based on reciprocal best BLAST hit). The differences in gene number and size are likely due to the different genome annotation strategies followed in each study and the lower contiguity of our assembly. Therefore, efforts to merge both assemblies and generate a reference version are warranted.

*In addition, the genomes of long-lived, slow growing *Lamellibrachia luymesii* (Li et al. 2019) and *Paraescarpia echinospica* (Sun et al. 2021) are from vestimentiferans from hydrocarbon seeps and therefore are from very different habitats than hydrothermal vents with short-living, fast growing *Riftia*. While I can understand that it is difficult to incorporate a new published paper when all analyses are already performed and the text in progress of writing, here clearly the path of downplaying was chosen and this is not correct. To simply state in the introduction that ‘our understanding of the genetic traits that sustain symbioses in Siboglinidae is scarce and currently limited to Vestimentifera’ is not enough to acknowledge three very different and detailed genome publications.*

RESPONSE: We have now expanded the introduction to reflect all the previous knowledge on the genomics of the bacterial symbiosis of Vestimentifera. In addition to the studies the reviewer mentioned, we also include the recent preprint on the genome of *Ridgeia piscesae* (<https://doi.org/10.1101/2022.08.16.504205>)

Overall, the bioinformatics part of the paper is really good. There are no major flaws, the pipeline follows a logical path and includes all the required steps for pre- and post-processing of PacBio and Illumina data.

RESPONSE: Many thanks.

In the lines 320-323, the authors insisted twice that immune-related genes are expanded in Vestimentifera, however, that is not the case in either the draft Riftia genome available in this paper, neither in previous publication. I suggest to rephrase the sentence to better describe the reality of the findings. The same applies for the conclusion paragraph (Line 329 - 331). The authors are not consistent throughout the paper.

RESPONSE: We have now rephrased all these sentences (lines 357 and 372–373). We have also adapted Figure 7 (the conclusion panel) to reflect that there are differences in immune-related gene contents within Vestimentifera.

*The authors state that *O. frankpressi*, in contrast to vestimentiferans and other annelids, show gene losses and expansions in key functional groups that support metabolic adaptations to its symbiotic lifestyles. This, however, is the case for every vestimentifera paper in the literature. Lamellibrachia, Paraescarpia, Riftia, all described gene families that are contracted and other that are expanded and linked this to the symbiosis and habitat. The authors also state that Vestimentifera experienced an event of gene family expansion in its last common ancestor (which is true and showed in the Supp. Figure 4B). However, this is not the same to say that Riftia, Paraescarpia and Oasisia STILL have an expanded repertoire of x,y and z. Their own Supp. Figure 4B shows that Riftia, Paraescarpia and Oasisia are losing way more genes than gaining and thus experiencing maybe reductive evolution in their genomes.*

RESPONSE: Thanks for pointing this out. We now clarify this point (lines 187–188 and 195–196).

*Overall, all this points to the authors' are trying too hard to show that *Osedax* is incredibly different from other systems, but in fact this has not been shown clearly. Instead, there might be more similarities than differences. The authors might consider to discuss this topic more openly.*

RESPONSE: In addition to the changes to address the previous points, we have toned down some of our previous statements about how different *O. frankpressi* is from Vestimentifera and focused on what is arguably different between the two lineages (e.g., genome size, GC content, number of genes and differences in gene repertoires). We also highlight similarities, such as in the developmental genetic toolbox.

*Concerning the symbionts – Also there is no overview provided in the introduction. Further, the authors state that 'we sequenced the hologenome of *Osedax frankpressi* Rouse, Goffredi & Vrijenhoek, 2004, as well as that of two vent dwelling Vestimentifera, *Oasisia alvinae* Jones, 1985 and *Riftia pachyptila* Jones, 1981'. They say that the "hologenome" is sequenced, but they don't cite where it has been published and by who. Also, here I would expect the citation of all genomes of endosymbionts including the most recent one of the Riftia from de Oliveira et al. 2022.*

RESPONSE: We have amended and clarified this section, adding the suggested references (lines 57–63).

*While the primary symbiont of *Osedax* is presented quite extensively, that of *Oasisia* is not. Also, I miss the annotation of *Oasisia* and *Riftia* symbiont in like Supplementary Table 12 done for *Osedax* symbiont.*

RESPONSE: We have now included a full annotation of *Oasisia*'s symbiont genome in Supplementary Table 10. For the endosymbiont of *R. pachyptila*, we relied on previously published assemblies and annotations.

*The authors use the word chemoautotrophic and chemotrophic as a synonym, not considering that a chemotroph can also be a heterotroph (as has been shown for a ciliate symbiont), hence a chemoheterotroph. Chemotrophs can be autotrophs, heterotrophs or mixotrophs. Fact is that *Endoriftia* shows many genes that support autotrophy (their function well supported since the very beginning of this research) as well as heterotrophy. The functioning as heterotroph symbiont has not been shown clearly yet, but since this is a genome paper the presence of genes for heterotrophy should be cited. I would really question the conclusion that the heterotrophic endosymbiont of *Osedax* is more metabolic versatile than *Endoriftia* and other vestimentiferan endosymbionts.*

RESPONSE: We have adapted the text to be more consistent with our use of chemoautotrophic and chemotrophic and provided additional background on the metabolic capabilities of the symbionts (lines 56–63). For continuity, we now use the term chemoautotrophic when referring to the Frenulata, *Sclerolinum* and Vestimentifera symbioses and heterotrophic for the symbiosis in *Osedax*. We now explicitly state that Vestimentifera endosymbionts are mixotrophs (line 58). We have also removed statements about the higher metabolic versatility of *Osedax*'s endosymbiont and refocused our discussion to our new results demonstrating that the Oceanospirillales symbiont contains unique metabolic properties that enable it to complement *Osedax*'s metabolism and that are not found in the Vestimentifera symbioses. We have also expanded our analysis of the symbiont secretion systems and revealed they contain numerous protein secretion genes with eukaryote-like protein domains, many of which are specific to their host lineages (as mentioned in response to Reviewer 1; lines 218–224, Supplementary Table 14).

As a more theoretical point – while the use of the term hologenome might be justified for symbioses with vertically transmitted, obligate symbionts without free-living populations, because selection acts on the holobiont only, this is clearly not the case for symbioses with horizontally transmitted symbionts – with host-associated and free-living populations and host stages that are symbiotic as well as without their symbiont during early development. Here, selection acts on multi-levels (as in siboglinids) and therefore the term hologenome is rejected by many researchers.

RESPONSE: We have removed the term hologenome from the manuscript.

Due to all the reasons outlined above I suggest a major revision.

Reviewer #4 (Remarks to the Author):

*There was a lot of great work that went into this manuscript and and it is a good look at the genomes of these animals. However there is one glaring omission. The genome of *Lamellibrachia* and the genomes of frenulate endosymbionts were not include or really even discussed in any of the analyses. This is a bit shocking as the authors are very much aware of this work and, more importantly the comparison between *Osedax* and *Lamellibrachia* is a better comparison as the water chemistry and environment of a whale fall and a seep are*

more similar than a whale fall and vents. This is a major oversight and should be corrected prior to publication. Granted the Lamellibrachia genome is a Illumina and not a PacBio genome but is it still good enough for many of the comparisons. Why the Li et al (2018) was excluded is not clear. Both of these papers (below) are very germane to the discussions in this paper.

RESPONSE: We now include the genome assembly and annotation of *Lamellibrachia luymesii* in all our analyses, including the study of immune genes and GPCRs, and have updated the figures accordingly. Following reviewer #3's suggestion, we have expanded the introduction to describe more extensively what was previously known about the genomic adaptations of Vestimentifera to their symbiotic life cycle (thus including and referring to the two suggested papers) (lines 50–63). In addition, we included the genome of the frenulate endosymbiont in our analyses and updated Supplementary Table 13 accordingly.

Also this paper suffers from the genomic shotgun approach meaning the manuscript tried to report on numerous genomic features that just are not relevant to the main story (ie Hox gene discussion, or why 28 genomes across metazoa are included). I will spare the other examples.

RESPONSE: Considering this and reviewer #1's concerns, we have shortened the section on developmental genes and DNA repair, moving the two main figures to Supplementary Information (new Supplementary Figures 11 and 12). We also clarify using 28 metazoan genomes to reconstruct gene families (lines 156–159; see point below).

The flow of the paper could be improved and several sentences are written awkwardly - especially in the Abstract. For example line 34 starts "endosymbionts ultimately..." and the reader does not know which endosymbionts are being discussed until the very end of the sentence. Or lines 126 and 129 which are very confusing on the repeat content of Capitella.

RESPONSE: We have rewritten the abstract and improved the flow of the paper by focusing the text on the molecular interactions between the host and endosymbiont. We have also expanded on what is already known about Vestimentifera in the Introduction, following reviewer #3 suggestion (lines 50–63). In addition, we have amended all sentences that were written awkwardly.

Line 144 it mentions that a PCA of genomes was conducted -- it is not at all clear what the input data was for the PCA -- what is being compared?

RESPONSE: The input data for the PCA plot is the number of orthologs for each species and gene family as generated by OrthoFinder. This is now in materials and methods (lines 611–613) and main text (lines 156–158).

The main text needs more on the quality of the genomes - N50, where they dovetailed or optically mapped? The low BUSCO for Osedax in concerning. In my experience with PacBio that signals a poor assembly. Yes they did find them by manually but this is a point of concern about quality.

RESPONSE: We now include the N50 values and number of scaffolds in the main text (lines 118–119). We now clarify that the relatively low number of BUSCO genes in the assembly is probably due to fast rates of molecular evolution (lines 128–131).

Li Y, Tassia MG, Waits DS, Bogantes VE, David KT, Halanych KM. 2019. Genomic adaptations to chemosymbiosis in the deep-sea seepdwelling tubeworm Lamellibrachia luymesii. BMC Biol. 17(1):91.

Li, Y., M. R. Liles, K. M. Halanych. 2018. Endosymbiont genomes yield clues of tube worm success. ISME Journal. 12:2785-2795. [/doi.org/10.1038/s41396-018-0220-z](https://doi.org/10.1038/s41396-018-0220-z)

RESPONSE: We include and reference these two studies in the revised manuscript extensively.

REVIEWERS' COMMENTS

Reviewer #2 (Remarks to the Author):

The authors have satisfactorily addressed my concerns.

In general, I think that this paper is excellent.

I have a few suggestions:

- Please consider mentioning the GTDB taxonomy of the symbionts.

L198 "Siboglinidae endosymbionts show unique metabolic traits" – this section title is quite vague, and the comparative basis for the uniqueness is not clear. Try to be more specific in the title. It is not surprising that the *Osedax* symbionts differ from the chemosynthetic ones, whereas the genome expansion compared to the free-living relatives is very interesting.

L234 – only oligopeptides? How about amino acids, etc.?

L270 "The glyoxylate cycle is a lineage-specific metabolic interaction of *Osedax* and its endosymbionts that could direct lipids in bones, the endosymbionts, and the root..." – please rephrase, it sounds like a definition of this cycle. Maybe: "Glyoxylate cycle may play a role in the metabolic interaction of *Osedax* and its endosymbionts by collectively converting bone lipids to carbohydrates, which are nearly absent in bones"

L236: Are you sure that tubeworm symbionts cannot produce B12? Some enzymes, including the cobalamin synthase, appear to be ubiquitous in them.

- Throughout the text – please check if italicizing *Oceanospirillales* is required by Nature Communications.

RESPONSE TO REVIEWERS' COMMENTS

Reviewer #2 (Remarks to the Author):

The authors have satisfactorily addressed my concerns.

In general, I think that this paper is excellent.

I have a few suggestions:

- Please consider mentioning the GTDB taxonomy of the symbionts.

RESPONSE: We now include this in the main text: "... we used our PacBio long-read data to assemble the genomes of the primary endosymbionts of *O. frankpressi* (Rs1 ribotype; **Genome Taxonomy Database accession number Rs1 sp000416275**) (Fig. 3a; Supplementary Data 3)..." (lines 200–202)

L198 "Siboglinidae endosymbionts show unique metabolic traits" – this section title is quite vague, and the comparative basis for the uniqueness is not clear. Try to be more specific in the title. It is not surprising that the Osedax symbionts differ from the chemosynthetic ones, whereas the genome expansion compared to the free-living relatives is very interesting.

RESPONSE: We have changed this section title to "The different genomic traits of Siboglinidae endosymbionts" (line 198).

L234 – only oligopeptides? How about amino acids, etc.?

RESPONSE: It is both amino acid and oligopeptide. We have amended this sentence (lines 234 and 235: "... and multiple sugar, amino acid and oligopeptide ATP-binding transporters (Supplementary Data 7).").

L270 "The glyoxylate cycle is a lineage-specific metabolic interaction of Osedax and its endosymbionts that could direct lipids in bones, the endosymbionts, and the root..." – please rephrase, it sounds like a definition of this cycle. Maybe: "Glyoxylate cycle may play a role in the metabolic interaction of Osedax and its endosymbionts by collectively converting bone lipids to carbohydrates, which are nearly absent in bones"

RESPONSE: We have changed the sentence as suggested (lines 271–273).

L236: Are you sure that tubeworm symbionts cannot produce B12? Some enzymes, including the cobalamin synthase, appear to be ubiquitous in them.

RESPONSE: The reviewer is correct, the endosymbionts can produce B12, but Frenulata and Vestimentifera endosymbionts lack one of the two alternative pathways. We have removed B12 from that sentence.

- Throughout the text – please check if italicizing Oceanospirillales is required by Nature Communications.

RESPONSE: There was no specific editorial comment on this; therefore, we have kept *Oceanospirillales* in italics.